# Calcium-stimulated disassembly of focal adhesions mediated by an ORP3/IQSec1 complex

Ryan S D'Souza[1], Jun Y Lim[1], Alper Turgut[1], Kelly Servage[2,3], Junmei Zhang[1], Kim Orth[2,3], Nisha G Sosale[4], Matthew J Lazzara[4], Jeremy Allegood[5], James E Casanova[1]*

[1]Department of Cell Biology, University of Virginia Health System, Charlottesville, United States; [2]Department of Molecular Biology, University of Texas Southwest Medical Center, Dallas, United States; [3]Howard Hughes Medical Institute, Dallas, United States; [4]Department of Chemical Engineering, University of Virginia, Charlottesville, United States; [5]Department of Biochemistry and Molecular Biology, Virginia Commonwealth University, Richmond, United States

**Abstract** Coordinated assembly and disassembly of integrin-mediated focal adhesions (FAs) is essential for cell migration. Many studies have shown that FA disassembly requires $Ca^{2+}$ influx, however our understanding of this process remains incomplete. Here, we show that $Ca^{2+}$ influx via STIM1/Orai1 calcium channels, which cluster near FAs, leads to activation of the GTPase Arf5 via the $Ca^{2+}$-activated GEF IQSec1, and that both IQSec1 and Arf5 activation are essential for adhesion disassembly. We further show that IQSec1 forms a complex with the lipid transfer protein ORP3, and that $Ca^{2+}$ influx triggers PKC-dependent translocation of this complex to ER/plasma membrane (PM) contact sites adjacent to FAs. In addition to allosterically activating IQSec1, ORP3 also extracts PI4P from the PM, in exchange for phosphatidylcholine. ORP3-mediated lipid exchange is also important for FA turnover. Together, these findings identify a new pathway that links calcium influx to FA turnover during cell migration.

*For correspondence:
jec9e@virginia.edu

## Introduction

Cell migration is central to a wide variety of physiological and pathophysiological processes. These include morphogenetic movements during embryogenesis, migration of inflammatory cells to sites of infection or injury, wound healing, tissue regeneration, neovascularization of solid tumors and the metastasis of tumor cells to distant anatomical sites. Migrating cells extend protrusions that form integrin-mediated contacts with the extracellular matrix, physically linking the ECM to the actin cytoskeleton. These nascent adhesions (focal contacts) mature to become larger, more stable focal adhesions (FAs) that serve as anchorage points for the actomyosin-dependent traction forces that pull the cell forward. During migration, mature adhesions disassemble at the rear of the cell, and internalized integrins are recycled to the cell front (*Huttenlocher and Horwitz, 2011*; *Webb et al., 2002*). While significant strides have been made in understanding how adhesions form, our understanding of how they disassemble remains fragmentary.

It is well established that $Ca^{2+}$ is a critical regulator of cell migration, at least in part by stimulating calpain-mediated cleavage of focal adhesion components (*Bhatt et al., 2002*; *Pettit and Fay, 1998*; *Wei et al., 2012*). Many cell surface receptors, including integrins, activate phospholipase Cγ (PLCγ) resulting in $IP_3$-mediated $Ca^{2+}$ release from ER stores. Depletion of ER $Ca^{2+}$ is complemented by an ensuing influx of extracellular $Ca^{2+}$ to the cytosol by Store Operated Calcium Entry (SOCE), mediated by the bipartite stromal interaction molecule 1 (STIM1)/Orai1 $Ca^{2+}$ channels (*Lewis, 2007*;

*Liou et al., 2005*; *Parekh and Penner, 1997*; *Prakriya et al., 2006*; *Roos et al., 2005*; *Vig et al., 2006*). In response to $Ca^{2+}$ depletion, the ER resident protein STIM1 translocates to the plasma membrane where it assembles with the $Ca^{2+}$ channel Orai1 at ER/plasma membrane contact sites (MCSs), leading to channel opening. Importantly, STIM1 and Orai1 are coordinately upregulated in many cancers, and pharmacological inhibition of SOCE has been shown to inhibit both migration in vitro and metastasis in vivo (*Chen et al., 2016*; *Chen et al., 2013*; *Yang et al., 2009*).

Our lab investigates the roles of Arf family GTPases and their regulators in cell migration. The Arfs are a family of small GTPases best known for their roles in vesicular transport (*D'Souza-Schorey and Chavrier, 2006*). Higher eukaryotes express six Arf isoforms divided into three classes, with Arf1, 2 and 3 in class I, Arf4 and 5 in class II and Arf6 the sole representative of class III. Most previous studies of adhesion and motility have focused on Arf6, which has been shown to regulate the recycling of integrins and other cargos from endosomal compartments to the plasma membrane (*Balasubramanian et al., 2007*; *Jovanovic et al., 2006*; *Powelka et al., 2004*).

Like other GTPases, Arfs require guanine nucleotide exchange factors (GEFs) to load them with GTP. The human genome contains 15 Arf GEFs (*Casanova, 2007*). Among these, the IQSec subfamily is characterized by the presence of an N-terminal calmodulin-binding IQ motif (*D'Souza and Casanova, 2016*). We have previously shown that IQSec2, which localizes to the post-synaptic densities of excitatory neurons, binds calmodulin and controls the trafficking of neurotransmitter receptors in a $Ca^{2+}$-dependent manner (*Myers et al., 2012*), but whether other isoforms are similarly regulated by $Ca^{2+}$ remains unknown.

IQSec1 (also referred to as BRAG2), which is ubiquitously expressed, has been linked to cell adhesion/migration in several contexts. Previously, we reported that depletion of IQSec1 leads to increased surface levels of β1-integrin, enhanced adhesion and a corresponding reduced rate of integrin endocytosis (*Dunphy et al., 2006*; *Moravec et al., 2012*). More recently, depletion of IQSec1 was shown to slow focal adhesion disassembly in keratinocytes (*Yue et al., 2014*) and also to inhibit vasculogenesis in vivo (*Manavski et al., 2014*). IQSec1 has also been implicated in the metastasis of breast (*Morishige et al., 2008*), lung (*Oka et al., 2014*), and pancreatic (*Xie et al., 2012*) cancers, and as well as melanoma (*Yoo et al., 2016*). While all of these studies highlight a vital role for IQSec1 in cell migration and invasion, we still do not know how IQSec1 mediates these processes.

Here we report that IQSec1 promotes focal adhesion disassembly through the activation of Arf5, which until now has been thought to function solely in the early secretory pathway (*Chun et al., 2008*; *Claude et al., 1999*; *Volpicelli-Daley et al., 2005*). We further show that depletion of Arf5, but not of Arf6, phenocopies IQSec1 depletion, leading to flattened, non-motile cells with enlarged focal adhesions and a reduced rate of adhesion turnover. Importantly, Arf5 activation is stimulated by $Ca^{2+}$ influx, and attenuated by inhibitors of SOCE. We also report the surprising discovery that IQSec1 activation is further stimulated by its association with oxysterol binding protein related protein 3 (ORP3), a lipid transfer protein previously reported to modulate cell adhesion (*Lehto et al., 2008*). We further show that ORP3 is recruited to ER/PM contact sites containing STIM1/Orai1 calcium channels, that cluster adjacent to focal adhesions. This recruitment is $Ca^{2+}$-dependent and requires PKC-mediated phosphorylation of ORP3. Interaction of ORP3 with IQSec1 is similarly $Ca^{2+}$/PKC-dependent, and inhibition of either $Ca^{2+}$ influx or PKC activity inhibits downstream activation of Arf5. In agreement with these observations, ORP3 depletion similarly inhibits FA disassembly and migration in both 2D and 3D matrices. We show that ORP3 binds and extracts PI4P from the plasma membrane, delivering it to the ER in exchange for a second lipid, which we identify here as phosphatidylcholine. A mutant of ORP3 lacking lipid exchange activity partially, but not completely, restores adhesion turnover in ORP3-depleted cells, suggesting that localized activation of Arf5 and insertion of PC into the plasma membrane are both important to promote adhesion disassembly. Taken together, our findings establish a spatial and mechanistic link between calcium influx at ER/PM contact sites and focal adhesion turnover mediated by ORP3 and IQSec1-activated Arf5.

## Results

### IQSec1 promotes focal adhesion disassembly

We have previously shown that depletion of IQSec1 in Hela cells leads to dramatic flattening of the cells and enhanced adhesion to fibronectin, due to accumulation of β1-integrins on the cell surface (*Dunphy et al., 2006*; *Moravec et al., 2012*). Similar observations have been reported in cultured endothelial cells (*Manavski et al., 2014*). Because previous reports have linked IQSec1 to metastatic breast cancer (*Morishige et al., 2008*), here we focused on the highly metastatic breast cancer cell line, MDA-MB-231.

Bioinformatics indicates that IQSec1 exists in a large number of splice variants (*Figure 1—figure supplement 1A*). By RT-PCR we determined that three of these variants are expressed in MDA-MB-231 cells, isoform A (NP_001127854.1), isoform X5 (XP_011532610.1) and isoform C (NP_001317548), the latter previously reported as GEP100 (*Someya et al., 2001*). Antibodies against IQSec1 recognize two bands of approximately 100 and 125 KDa in cell lysates, whose expression is reduced upon shRNA-mediated knockdown of IQSec1 using hairpins directed at the shared core sequence (*Figure 1—figure supplement 1B, C*).

As previously described in other cell types, silencing of IQSec1 in MDA-MB-231 cells yielded a dramatically flattened phenotype relative to controls (*Figure 1A*). Close inspection of IQSec1 depleted cells revealed a significant increase in the size (*Figure 1B–C*) and β1-integrin content (*Figure 1D*) of focal adhesions. A similar increase in adhesion size was also observed upon IQSec1 depletion in the invasive fibrosarcoma cell line HT1080, (*Figure 1—figure supplement 1D*).

Unlike control cells, which are highly motile (*Figure 1*; *Video 1*), IQSec1-depleted cells appeared to be fixed in place and completely non-motile (*Figure 1*; *Video 2*). Consistent with this, quantification of focal adhesion dynamics indicated that the rate of adhesion disassembly was significantly decreased relative to mock-depleted cells (*Figure 1E*). Note that the broad distribution in rates reflects distinct populations of adhesions that turn over at significantly different rates. Moreover, the software used for analysis of adhesion dynamics (*Berginski et al., 2011*) only measures adhesions that assemble and disassemble over the time course of imaging (60 min). Static adhesions, which are abundant in IQSec1-depleted cells, are therefore under-represented in this analysis. Importantly, disassembly rate was restored to near control levels by re-expression of wild-type IQSec1 isoform A/ NP_001127854.1 or isoform C/NP_001317548, but not a catalytically inactive mutant of isoform A (E606K) (*Figure 1E*). Interestingly, neither isoform B/NP_ 055684.3 nor isoform X5/XP_011532610.1 significantly rescued FA turnover, suggesting that the alternatively spliced N-terminus in these variants affects their function. This analysis also revealed a significant decrease in the rate of FA assembly (*Figure 1F*). We attribute this to the accumulation of integrins in enlarged, static adhesions, depleting the pool available for formation of new contacts. Consistent with the reduced rate of adhesion turnover, depletion of IQSec1 also dramatically impaired migration of MDA-MB-231 cells through Matrigel in a 3D transwell migration assay (*Figure 1G*), in agreement with an earlier report (*Morishige et al., 2008*).

Lastly, we used a more physiologically relevant assay to measure cell migration in 3D collagen gels. Multicellular tumor spheroids generated by growing MDA-MB 231 cells in low-adhesion U-bottom wells, were embedded in collagen gels and imaged live for 18–24 hr. While migration of cells from control spheroids was clearly observed, essentially no migration was observed from spheroids generated from IQSec1 depleted cells (*Figure 1H–I*). Taken together, these results show that IQSec1 modulates 2D as well as 3D cell migration and invasion.

### IQSec1 activates Arf5 to promote FA disassembly

We have previously shown that IQSec1 activates both Arf5 and Arf6 in HeLa cells (*Moravec et al., 2012*). To date, limited studies of Arf5 have focused on its roles early in the secretory pathway, whereas Arf6 has been shown to act at the plasma membrane and in endosomal recycling pathways (*Balasubramanian et al., 2007*; *Jovanovic et al., 2006*; *Oh and Santy, 2010*; *Powelka et al., 2004*). As we had observed in Hela cells (*Moravec et al., 2012*), knockdown of Arf6 in MDA-MB-231 cells did not phenocopy IQSec1 knockdown; Arf6-depleted MDA-MB-231 cells were actually less well spread than controls (*Figure 2—figure supplement 1A,C*). Conversely, depletion of Arf5 in these cells led to dramatic cell flattening (*Figure 2—figure supplement 1B,C*), a significant increase

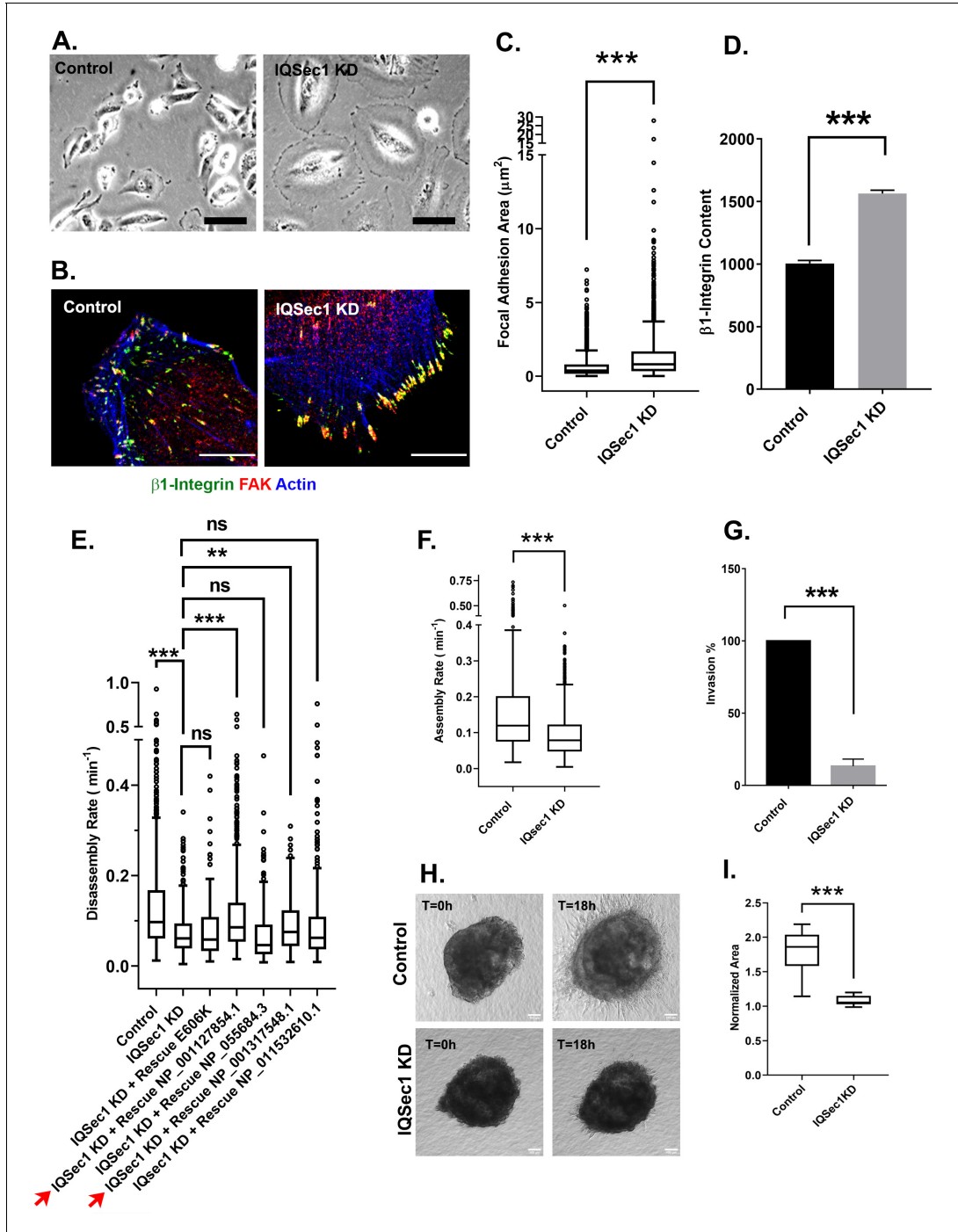

**Figure 1.** IQSec1 promotes focal adhesion disassembly in MDA-MB-231 cells. (**A**) Bright field images of control and IQSec1 depleted cells, Bar = 20 μm. (**B**) Control and IQSec1 depleted cells, fixed and stained for β1-integrin (green), FAK (red) and actin (blue), Bar = 10 μm. (**C**) *Quantitation of FA size from control and IQSec1-depleted cells.* Cells were fixed and stained for vinculin as a marker for FAs; N for control = 2487 and for IQSec1 KD = 1805 FAs were measured. 30 cells were analyzed per group, compiled from three independent experiments. (**D**) *Quantitation of β1-integrin fluorescence in FAs from control and IQSec1 depleted cells.* N for control = 1938 control and for IQSec1 KD = 827 FAs. 5 cells were analyzed per group. (**E**) *Rescue of IQSec1 depletion by IQSec1 variants.* FA disassembly rates were measured in MDA-MB-231 cells expressing GFP-paxillin (See **Videos 1** and **2**). Cells were depleted of endogenous IQSec1 using a hairpin directed against a sequence shared by all isoforms, then transfected with constructs encoding each individual isoform, engineered to lack the shRNA target sequence. E606K is a catalytically inactive mutant of isoform C (NP_001127854.1). Both NP_001127854.1 and NP_001317548.1 (red arrows) increase FA turnover, although the rescue never reaches control levels. Both NP_055684.3 and NP_001317548.1 are similar to the catalytically inactive E606K mutant and do not rescue. N for control = 1280, IQSec1 KD = 1150, IQSec1 KD+ WT Rescue NP_001127854.1 = 817, IQSec1 KD+ Rescue E606K = 602, IQSec1 KD + Rescue NP_055684.3 = 336, IQSec1 KD + Rescue

*Figure 1 continued on next page*

*Figure 1 continued*

NP_001317548.1 = 447, IQSec1 KD + Rescue NP_001317548.1 = 561 FAs. Data were collected from 20 control and 20 IQSec1-KD cells and 12 cells per group for others. Data were compiled from three independent experiments. (F) *FA assembly rates in cells expressing GFP-paxillin.* N for control = 689 and for IQSec1 KD = 763 FAs. Data were collected from 12 cells per group spanning three independent experiments. (G) *Migration of control or IQSec1-depleted cells through a matrigel plug.* Migration was measured as described in Methods. Data were compiled from three independent experiments, each done in triplicate. (H) *Migration of cells out of tumor spheroids embedded in collagen gels.* Stills from time lapse movies of tumor spheroids cultured in 3D collagen gels, imaged for 18 hr. Bar = 100 µm. (I) *Quantitation of cell migration out of spheroids after 18* hr. Measurement details are described in Methods. Data were collected from 12 spheroids from each group. '*' indicates p<0.05, '**' indicates p<0.001, and '***' indicates p<0.0001. The same annotation is used in all following figures.

The online version of this article includes the following figure supplement(s) for figure 1:

**Figure supplement 1.** IQSec1 splice variants, expression in MDA-MB-231 cells and knockdown efficiency.

in focal adhesion area (*Figure 2—figure supplement 1D*) with higher β1-integrin content (*Figure 2—figure supplement 1F*), and reduced disassembly rate (*Figure 2—figure supplement 1G*). Similar increases in focal adhesion area were observed in HT1080 cells (*Figure 2—figure supplement 1E*). Together, these results strongly suggest that Arf5 but not Arf6 is essential for focal adhesion turnover.

To provide additional support for this observation, we sought to bypass the IQSec1 deficiency using GEF-independent Arfs; rapid cycling Arf mutants that have a lower affinity for nucleotide, enabling them to undergo rapid, GEF-independent nucleotide exchange. Given the high cytosolic ratio of GTP/GDP, these mutants exist primarily in the active, GTP bound state but unlike traditional constitutively active mutants they also retain the ability to hydrolyze GTP, and therefore do not sequester effector proteins (*Santy, 2002*). As shown in *Figure 2B*, expression of rapid cycling Arf6 failed to restore FA turnover in IQSec1-depleted cells. In contrast, expression of rapid cycling Arf5 largely restored adhesion turnover (*Figure 2B*). Like IQSec1, knockdown of Arf5 substantially impaired migration in the matrigel migration assay (*Figure 2C*). Together, these data suggest that activation of Arf5, not Arf6, by IQSec1 mediates focal adhesion disassembly in migrating cells.

## Arf5 is activated in response to calcium influx

It is well established that $Ca^{2+}$ influx is required for focal adhesion disassembly in migrating cells (*Yang et al., 2009*). We have previously shown that the neuronal IQSec isoform IQSec2/BRAG1, which localizes to post-synaptic densities of excitatory neurons, is stimulated by $Ca^{2+}$ influx via NMDA receptors (*Myers et al., 2012*). To determine if the ubiquitously expressed IQSec1 responds similarly to $Ca^{2+}$ in non-neuronal cells, we monitored Arf5 activity using a pulldown assay previously developed by our laboratory (*Santy and Casanova, 2001*). For this purpose, cells expressing Arf5 along with either IQSec1 or the catalytically inactive E606K mutant, were treated with the SERCA pump inhibitor thapsigargin for 30 min in the presence of calcium. Thapsigargin depletes $Ca^{2+}$ from the ER by blocking the SERCA-mediated uptake of cytosolic $Ca^{2+}$ into the ER lumen (*Lytton et al., 1991*). This causes translocation of STIM1 to the PM where it binds to and activates PM localized Orai1 channels, resulting in an influx of calcium into the cell (*Liou et al., 2005*) As shown in *Figure 2D* - E, thapsigargin-induced calcium influx increased active Arf5 by ~50% over basal levels in cells expressing WT IQSec1, whereas cells expressing the IQSec1 E606K mutant did not show a significant change in active Arf5

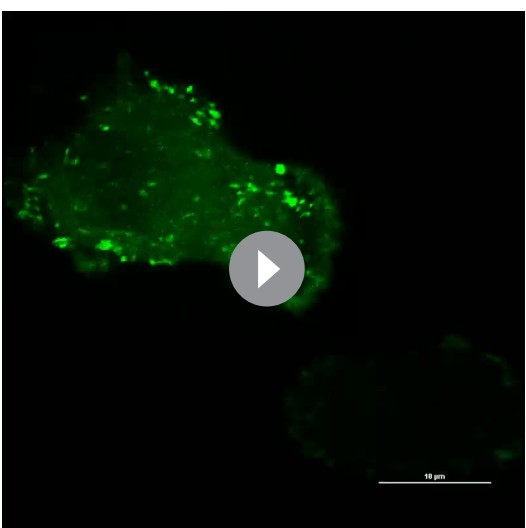

**Video 1.** MDA-MB-231 cell expressing GFP-paxillin, imaged for 60 min at a rate of 1 frame every 15s. Bar = 10 µm.
https://elifesciences.org/articles/54113#video1

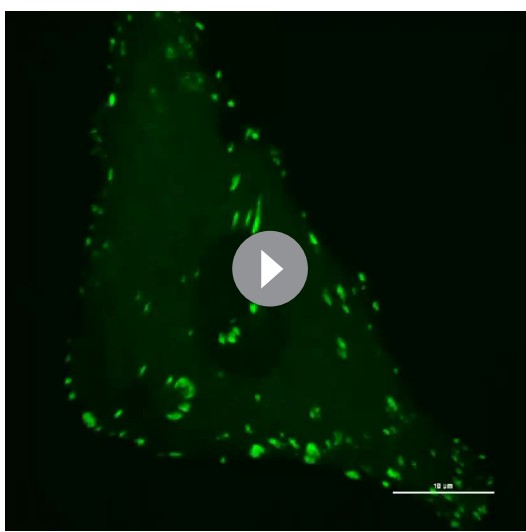

**Video 2.** MDA-MB-231 cell depleted of IQSec1 and expressing GFP-paxillin, imaged for 60 min at a rate of 1 frame every 15 s. Bar = 10 μm.
https://elifesciences.org/articles/54113#video2

levels. In contrast, Arf5 activation is inhibited by ~60% by the selective Orai1 channel blocker, BTP2 (*Figure 2F–G*), indicating that Arf5 activation is indeed modulated by calcium influx. Residual activity in the presence of BTP2 presumably represents Arf5 in the early secretory pathway, where it is activated by a different GEF, GBF1 (*Chun et al., 2008*; *Claude et al., 1999*; *Volpicelli-Daley et al., 2005*). As expected, we also observed an increase in FA size in MDA-MB-231 cells depleted of STIM1 (*Figure 2H*). Similarly, no significant migration was observed out of spheroids generated from STIM1 depleted cells in 3D collagen gels (*Figure 2I*).

Interestingly, we noted that GFP-STIM1 formed puncta that clustered near focal adhesions at the trailing edge of migrating cells (*Figure 2*; *Video 3*). As expected, formation of these puncta, which represent sites of contact between the ER and PM, is robustly stimulated by thapsigargin (*Figure 2—figure supplement 1H,I*). Importantly, this occurs similarly in cells depleted of either Arf5 or IQSec1, indicating that STIM1/Orai1 act upstream of IQSec1 (*Figure 2—figure supplement 1H,I*).

## IQSec1 associates with the lipid transfer protein ORP3

To identify proteins that may cooperate with IQSec1 to mediate FA turnover, GFP-IQSec1 was expressed in HEK293 cells and interacting proteins were identified by affinity precipitation followed by protein mass spectrometry (*Figure 3—source data 1*). Among the proteins identified were clathrin heavy chain (CLTC) and all four subunits of the AP-2 adaptor complex (AP2A1, AP2B1, AP2M1, AP2S1) which we have previously shown to interact with IQSec1 (*Moravec et al., 2012*). Consistent with the presence of a C-terminal PDZ motif in IQSec1, we also identified several proteins containing PDZ domains, including DLG1, PTPN13, SCRB1, ZO-2, MAGI3, PDLIM1/7, GIPC1 and GOPC.

Among other interesting hits was oxysterol binding protein related protein 3 (ORP3). ORP3 is a member of a family of proteins that mediate the non-vesicular transfer of lipids between the ER and other organelles at membrane contact sites (*Olkkonen, 2015*). Structurally, ORP3 contains an N-terminal PH domain, two centrally located FFAT motifs that mediate interaction with the ER resident proteins VAP-A and VAP-B, and a C-terminal lipid transfer domain (ORD domain). ORP3 caught our attention for three reasons: 1) the closely related protein OSBP is recruited to the TGN through an interaction of its PH domain with Arf1 (*Mesmin et al., 2013*), 2) ORP3 is unique among the ORPs in its ability to modulate cell adhesion and 3) overexpression of ORP3 was found to impair cell spreading and reduce activation of β1-integrins, while silencing of ORP3 enhances cell spreading (*Lehto et al., 2008*).

We first confirmed by immunoblotting that ORP3 co-precipitates with IQSec1 from lysates of cotransfected cells (*Figure 3A*). Importantly, silencing of ORP3 in MDA-MB-231 cells (*Figure 3B–C*) or HT-1080 cells (*Figure 3—figure supplement 1*) phenocopied knockdowns of IQSec1 and of Arf5, leading to flattened cells with enlarged focal adhesions and a reduced rate of FA turnover (*Figure 3D–E*). Moreover, like IQSec1-depleted cells, cells depleted of ORP3 were dramatically inhibited in their ability to migrate in 3D through Matrigel in the transwell migration assay (*Figure 3F*) and from spheroids embedded in collagen gels (*Figure 3G*).

Intriguingly, live imaging of MDA-MB-231 cells expressing GFP-ORP3 revealed that it formed puncta clustered in retractions where focal adhesion disassembly occurs, and often in close apposition to paxillin-labeled adhesions prior to disassembly (*Figure 3H–J*, *Video 4*). Quantitation of ORP3 localization confirmed that ORP3 puncta were enriched in retracting regions of cells (*Figure 3I*) and

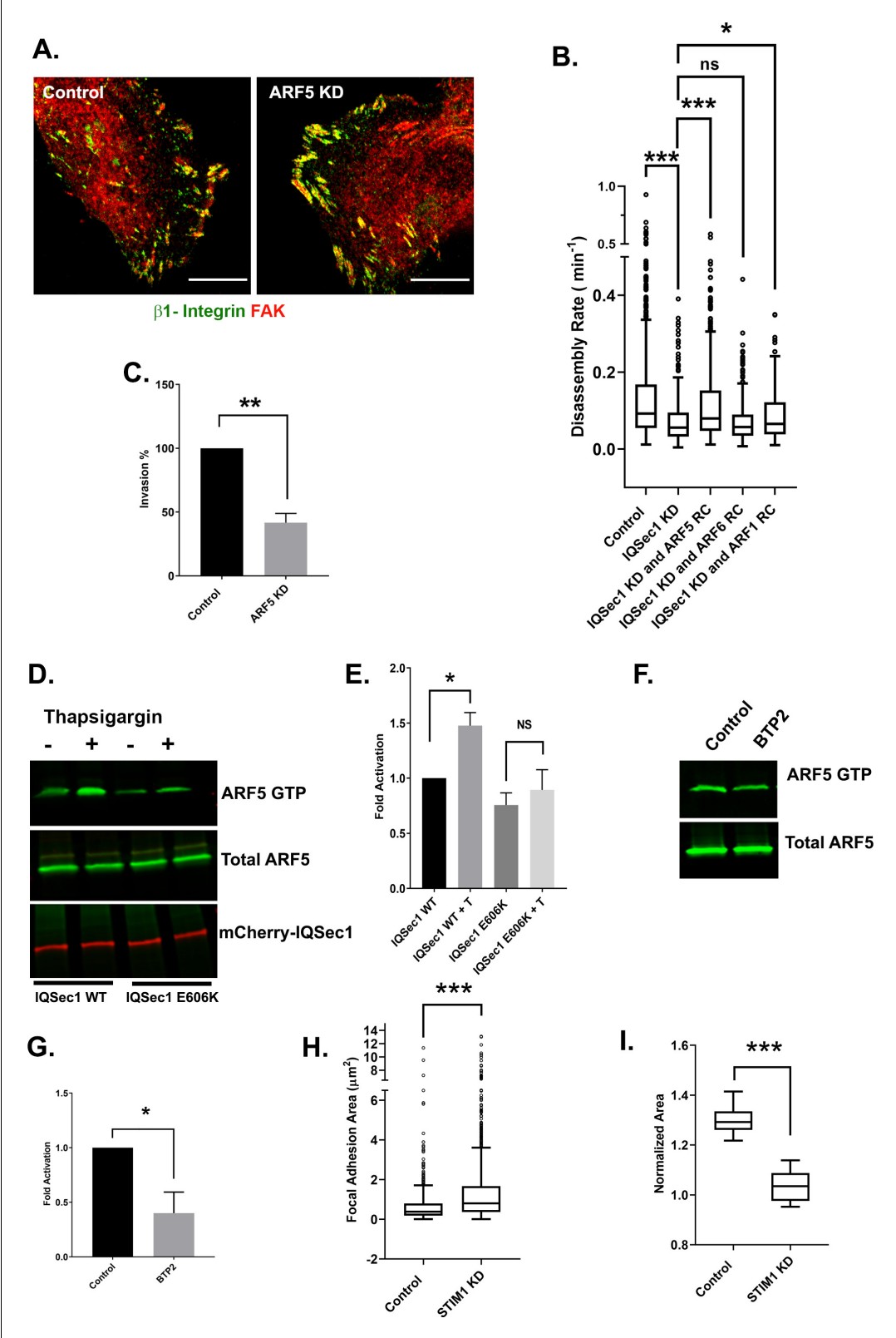

**Figure 2.** IQSec1 activates ARF5 to promote FA disassembly. (**A**) *Arf5 depletion leads to increased focal adhesion size.* Control (left) and Arf5-depleted cells (right) were stained for endogenous β1-integrin (green) and FAK (red). Bar = 10 µm. (**B**) *Expression of rapid cycling Arf5, but not Arf6 or Arf1, enhances adhesion turnover in IQSec1-depleted cells.* IQSec1-depleted cells were co-transfected with plasmids encoding GFP-paxillin and rapid cycling (RC) mutants of Arf6, Arf5 or Arf1. N for control = 1203, IQSec1 KD = 670, IQSec1 KD + Arf5 RC = 1087, IQSec1 KD + Arf6 RC = 612, IQSec1 KD + Arf1

*Figure 2 continued on next page*

*Figure 2 continued*

RC = 798 FAs. Data were collected from 15 cells per group, spanning three independent experiments. (**C**) *Migration of control vs. Arf5-depleted cells through a Matrigel plug.* Data were compiled from three independent experiments, each performed in triplicate. (**D** and **E**) *Pulldown assay for Arf5 activity.* HEK293 cells expressing Arf5-HA and either WT mCherry-IQSec1 or catalytically inactive mCherry-IQSec1[E606K] were cultured in serum and calcium free media for 3 hr, then stimulated with thapsigargin [T] (1 µM) in the presence of calcium (1 mM) for 30 min. Cells lysates were then incubated with GST-GGA3 beads to precipitate active, GTP-bound Arf5. (**D**) *A representative blot showing enhanced activation of Arf5 by thapsigargin in the presence of IQSec1 WT, but not IQSec1[E606K]* (**E**) *Quantitation of results from three independent experiments shown in 'D'.* (**F**) *Representative blot showing decreased Arf5 activity in cells treated with the Orai1 inhibitor BTP2 (25 µM).* (**G**) *Quantitation of results from three independent experiments shown in 'F'.* (**H**) *Quantitation of FA size from control and STIM1 depleted cells.* Cells were fixed and stained for endogenous vinculin. N for control = 848 and for STIM1 KD = 1603 FA. 30 cells were analyzed per group. (**I**) *Quantitation of cell migration out of spheroids derived from control and STIM1 depleted MDA-MB-231 cells.* Data were collected from eight spheroids for each group.

The online version of this article includes the following figure supplement(s) for figure 2:

**Figure supplement 1.** Knockdown of Arf5 but not Arf6 phenocopies IQSec1 depletion.

---

that nearly 100% of disassembling adhesions contained adjacent ORP3 puncta, while only ~25% of newly formed adhesions in protrusions did so (*Figure 3J*).

## ORP3 is recruited to ER/PM Contacts in Response to Calcium Influx

Other members of the ORP family have been shown to function at ER-membrane contact sites, and previous studies indicate that ORP3 puncta make contacts between the ER and plasma membrane. Surprisingly, we found that ORP3 not only forms ER/PM contacts but that it also colocalizes robustly with STIM1 at these sites (*Figure 4A–B*, *Video 5*).

At steady state, a relatively small number of ORP3 puncta was observed. However, treatment of cells with thapsigargin led to a dramatic increase in the number and intensity of ORP3 puncta that localized at contacts all over the ventral surface, often in linear arrays. The intensity of these puncta reached a peak at ~4.5 min after stimulation and declined over time (*Figure 4C–D*). Colocalization of ORP3 with STIM1 also increased, (*Figure 4A–B*), as well as with another marker of ER/PM contacts, the extended synaptotagmin E-Syt1 (*Giordano et al., 2013*; *Figure 4—figure supplement 1A*). In contrast, treatment of cells with the Orai1 blocker BTP2 or knockdown of STIM1 led to a significant loss of PM puncta and a redistribution of ORP3 to the cytoplasm, indicating that recruitment of ORP3 to the PM requires calcium, even at steady state (*Figure 4E–G*). Treating cells with EGF, which elevates intracellular calcium through activation of PLCγ, also resulted in the co-recruitment of ORP3 and STIM1 to the PM, albeit more slowly and less dramatically compared to thapsigargin treatment (*Figure 4—figure supplement 1B and B'*).

To determine if calcium modulates the interaction of ORP3 with IQSec1, cells were treated with either thapsigargin or BTP2, and the interaction monitored by co-precipitation. As shown in *Figure 4H and I*, a moderate basal level of association is significantly increased upon treatment of cells with thapsigargin. In contrast, co-precipitation is reduced in cells pretreated with BTP2. Together, these data suggest that the interaction between ORP3 and IQSec1 is modulated by intracellular calcium.

## ORP3 recruitment to the PM requires PI(4,5)P₂

ORP3 has been reported to localize to the plasma membrane via an interaction between its N-terminal PH domain and the 3-phosphoinositides PI(3,4)P$_2$ and PI(3,4,5)P$_3$ (*Lehto et al., 2005*). In agreement with that earlier study, we found that the ORP3 PH-domain is required for recruitment to the PM in response to thapsigargin (*Figure 5 - Figure 4—figure supplement 1C*). However, recruitment was not sensitive to the PI3-kinase inhibitor wortmannin, indicating that 3-phosphorylated phosphoinositides are not essential for PM targeting (*Figure 5 - Figure 4—*

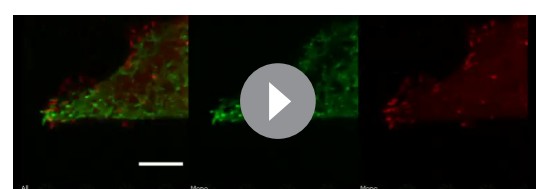

**Video 3.** Movie showing GFP-STIM1 making contact with Ds-Red-paxillin-labelled focal adhesions. Cell was imaged live for 30 min at a rate of 1 frame every 15 s. Bar = 2.5 µm.
https://elifesciences.org/articles/54113#video3

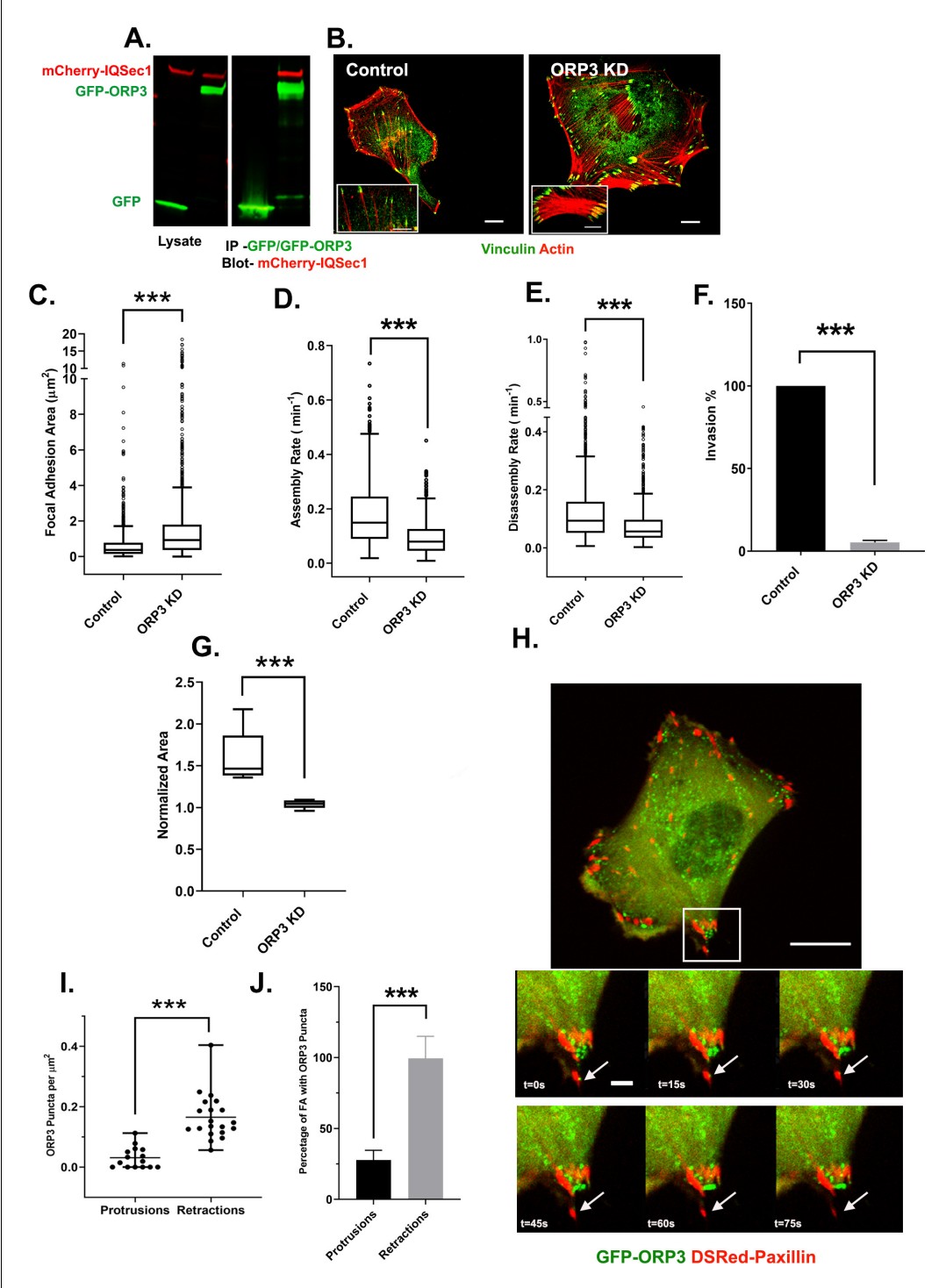

**Figure 3.** IQSec1 associates with ORP3 and ORP3 knockdown phenocopies IQSec1 knockdown. (**A**) Lysates from cells co-expressing mCherry-IQSec1 and either unfused GFP or GFP-ORP3 (left) were immunoprecipitated with GFP-TRAP beads (Chromotek) and probed by western blotting for GFP and mCherry (right). (**B**) MDA-MB-231 cells were either mock-depleted or depleted of ORP3 then fixed and stained for endogenous vinculin (green) and actin (red), Bar = 10 μm. (**C**) Quantification of FA size from control or ORP3 depleted cells, using vinculin as a marker for FAs, N for control = 1385 and for ORP3 KD = 1396 focal adhesions. Data were collected from 30 cells for each condition, spanning three experiments. (**D**) FA assembly rates in cells expressing GFP-paxillin. N for control = 804 and for ORP3 KD = 709 FA. Data were collected from 15 cells per group spanning three independent experiments. (**E**) FA disassembly rates in cells expressing GFP-paxillin. N for control = 1301 and for ORP3 KD = 1504 FAs. Data were collected from 15 cells per group spanning three independent experiments (**F**) Knockdown of ORP3 inhibits migration through a Matrigel plug. Data were compiled from three independent experiments, each performed in triplicate. (**G**) Quantification of cell migration out of spheroids after 22 hr in 3D collagen gels. Data

*Figure 3 continued on next page*

*Figure 3 continued*

were collected from seven spheroids from each group. (H) Time series from *Video 4*, showing an MDA-MB-231 cell co-expressing GFP-ORP3 and DsRed-paxillin. Panels show clustering of ORP3 puncta adjacent to disassembling FAs. Arrows indicate a single ORP3 punctum in close apposition with a paxillin-labelled FA before it disassembles. Bar on whole cell = 10 μm and on time series panel = 1 μm. (I) Quantification of the number of ORP3 puncta in protrusions (15 areas) and retractions (19 areas) from a total of 13 cells analyzed, compiled from three independent experiments. Protrusions and retractions were identified as specified in Methods (J) Prevalence of FAs with adjacent ORP3 puncta in protrusions vs. retracting regions of migrating cells. Proximity of puncta to adhesions was measured as described in Methods. Data were collected from 352 FAs in protrusions and 300 FAs in retractions.

The online version of this article includes the following source data and figure supplement(s) for figure 3:

**Source data 1.** Mass spectroscopy of IQSec1-interacting proteins- data underlying *Figure 3A*.
**Figure supplement 1.** Knockdown of ORP3 also increases focal adhesion area in HT1080 cells.

*figure supplement 1C and D*). Similar results were obtained with the PI3K inhibitor LY294002 (data not shown). Depletion of PM-associated PI4P using the PI4KIIIα inhibitor GSK-A1 (*Bojjireddy et al., 2014*) also did not impair formation of ORP3 puncta in response to thapsigargin treatment (*Figure 5 - Figure 4—figure supplement 1E*), indicating that recruitment is not dependent upon PI4P. PI4P levels were monitored using the well-characterized probe GFP-P4M2X (*Hammond et al., 2014*). We note that GSK-A1 treatment results in loss of PI4P from the PM and redistribution of the probe to intracellular compartments. Additionally, depletion of PI4P with GSK-A1 did not impair the thapsigargin induced recruitment of ORP3Δ$^{ORD}$, a mutant lacking the ORD domain (*Figure 5 - Figure 4—figure supplement 1F*), indicating that lipid binding by the ORD domain is not essential for membrane targeting.

In contrast, we found that ORP3 recruitment was dramatically inhibited by optogenetic depletion of plasma membrane PI(4,5)P$_2$ (*Figure 5A–B*). For this purpose, MDA-MB-231 cells were transfected with a construct encoding the inositol 5'-phosphatase domain of OCRL fused to the photolyase homology domain of CRY2 and CRY2 binding domain (CINB) fused to the plasma membrane targeting motif CAAX. Blue light illumination promotes binding of CRY2 with CINB, targeting the 5'-phosphatase domain to the PM and depleting PI(4,5)P$_2$ (*Idevall-Hagren et al., 2012*).

A complication of this experiment is that PI(4,5)P$_2$ is required for Orai1 channel function (*Ercan et al., 2009*; *Liou et al., 2007*; *Walsh et al., 2010*). To circumvent this problem, we treated cells with the phorbol ester PMA, which has been previously shown to promote translocation of ORP3 to the PM (*Weber-Boyvat et al., 2015*). As expected, incubation with PMA resulted in rapid recruitment of ORP3 to PM puncta in control cells (*Figure 5A–B*). In contrast, optogenetic depletion of PI(4,5)P$_2$ dramatically inhibited the recruitment of ORP3 to the PM (*Figure 5A–B*). Interestingly, optogenetic depletion of PI(4,5)P$_2$ also resulted in an increase in focal adhesion size (*Figure 5C* and *Figure 4—figure supplement 1G*). Taken together, these results indicate that the recruitment of ORP3 to the PM is mediated by the association of its PH domain with PI(4,5)P$_2$.

## ORP3 acts upstream of IQSec1 to stimulate Arf5 activation

Previous studies have shown that interaction of the closely related protein OSBP with the TGN is mediated by a coincidence detection mechanism in which its N-terminal PH domain binds simultaneously to both PI4P and Arf1 (*Mesmin et al., 2013*). This suggested that recruitment of ORP3 to the PM might be mediated by a similar

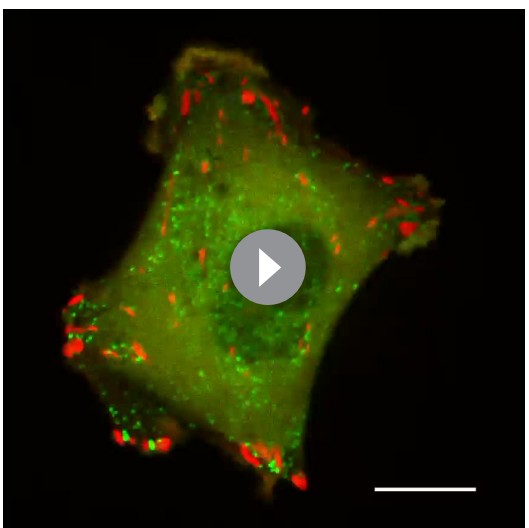

**Video 4.** MDA-MB-231 cell expressing GFP-ORP3 and Ds-Red-Paxillin was imaged live for 30 min at a rate of 1 frame every 15 s. Notice the accumulation of ORP3 puncta around FAs at retracting edges of the cell. Bar = 10 μm.
https://elifesciences.org/articles/54113#video4

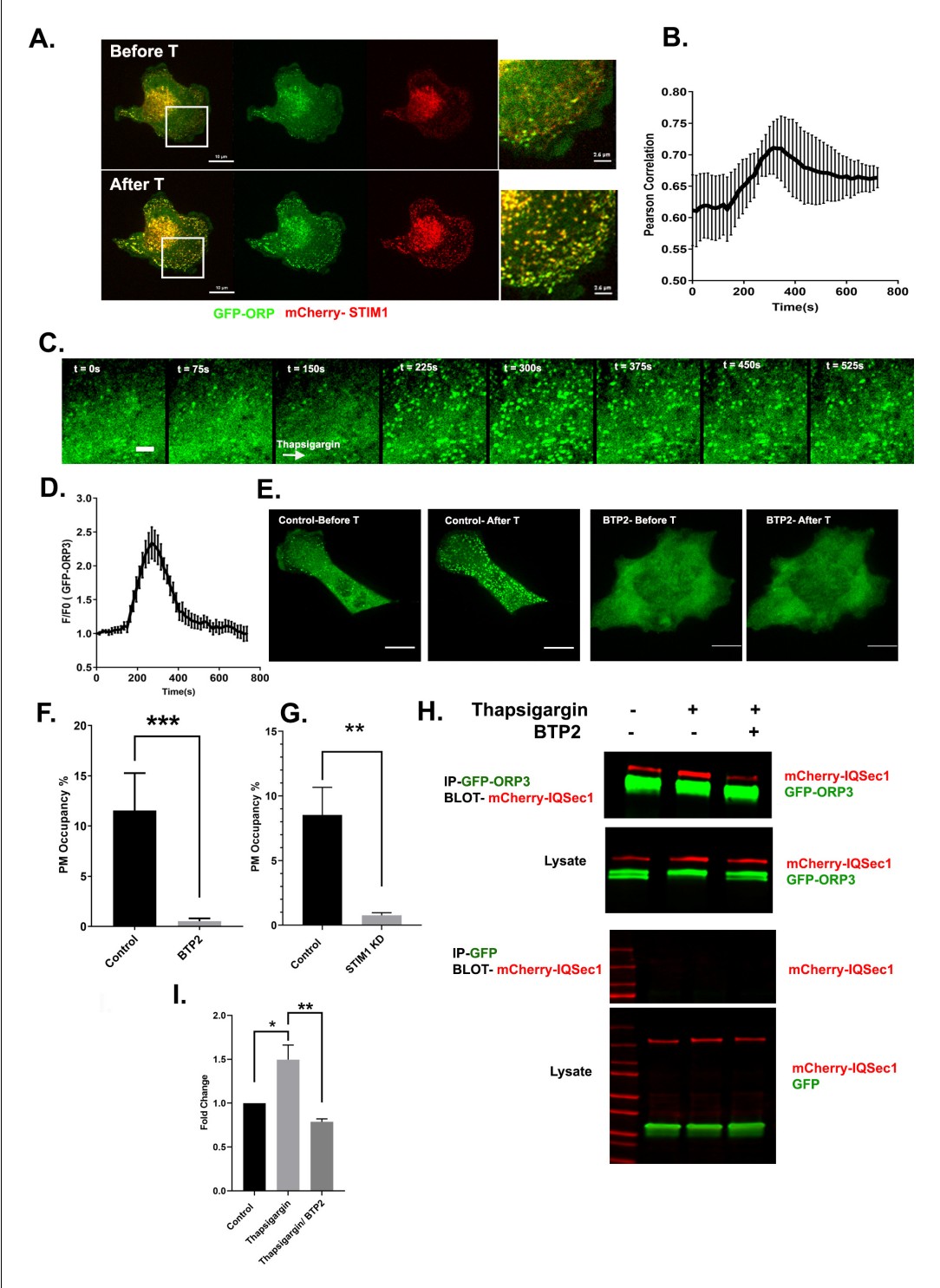

**Figure 4.** ORP3 is recruited to STIM1-positive ER/PM contacts in response to calcium influx. (**A**) Still images from live cells (*Video 5*) co-expressing GFP-ORP3 and mCherry-STIM1, before and after thapsigargin (1 μM) treatment. Bar = 10 μm. Magnified view of the boxed area ia shown at right. Bar = 2.5 μm (**B**) Quantification of ORP3 colocalization with STIM1 in response to thapsigargin, added 2.5 mins after the start of the movie. Data were collected from 10 regions of interest from three independent experiments. (**C**) Time series showing an MDA-MB-231 cell expressing GFP-ORP3, imaged before and after thapsigargin (1 μM) treatment (arrow indicates time of addition) showing robust recruitment of ORP3 to the PM after treatment with thapsigargin. Bar = 5 μm. (**D**) Quantification of ORP3 fluorescence at the PM, after thapsigargin treatment. Data were collected from 10 regions of interest from three independent experiments. (**E**) Live cells expressing GFP-ORP3 were pre-treated with either DMSO or with BTP2 (25 μM) for 2 hr, then treated with thapsigargin (1 μM) for 5 min. Bar = 10 μm. (**F**) Quantification of GFP-ORP3 recruitment to the PM after thapsigargin treatment in the
*Figure 4 continued on next page*

*Figure 4 continued*

presence and absence of BTP2. PM occupancy was calculated as described in Methods. Data were collected from 15 cells per group spanning three experiments. (G) Recruitment of GFP-ORP3 to the PM after thapsigargin treatment in control vs. STIM1 depleted cells. Data were collected from 13 cells per group, spanning three experiments. (H) Interaction between IQSec1 and ORP3 is stimulated by SOCE and inhibited when calcium influx is blocked by BTP2. Note the shift in ORP3 mobility upon thapsigargin treatment. Data from three independent experiments were quantified and presented in (I).

The online version of this article includes the following figure supplement(s) for figure 4:

**Figure supplement 1.** Recruitment of ORP3 to the plasma membrane.

mechanism in which its PH domain would bind simultaneously to PI(4,5)P$_2$ and Arf5. To test this hypothesis, we measured the Ca$^{2+}$-mediated PM recruitment of ORP3 in cells depleted of either IQSec1or Arf5. Surprisingly, we found that ORP3 was robustly recruited to PM puncta in IQSec1- or Arf5-depleted cells (*Figure 5D and E*), indicating that Arf5 is not essential for ORP3 recruitment to the PM.

We next tested the possibility that ORP3 might act upstream of IQSec1 to modulate its catalytic activity. For this purpose, we compared Arf5 activation in control cells and ORP3 depleted cells, using the pulldown assay described above. As shown in *Figure 5F*, silencing of ORP3 led to a > 50% reduction in Arf5 activity (quantified in *Figure 5G*), indicating that ORP3 is an important regulator of IQSec1 catalytic activity.

## Interaction of ORP3 with IQSec1

In principle, ORP3 could stimulate IQSec1 allosterically through a direct interaction, or indirectly, by delivering lipids to the PM that are essential for IQSec1 activity (*Aizel et al., 2013*). The IQ motif in IQSec proteins resides in an extended N-terminal region (*Figure 6—figure supplement 1A*). Crystal structures exist of the IQSec1 catalytic domain in tandem with its C-terminally adjacent PH domain, but these were generated from truncated constructs lacking the N-terminal domain (*Aizel et al., 2013*). Importantly, these constructs are constitutively active in in vitro assays (*Aizel et al., 2013*). This, together with our previous observation that calmodulin binding modulates catalytic activity in IQSec2 (*Myers et al., 2012*), suggests that the IQSec1 N-terminus may serve as a regulatory domain. Predictive modeling of the N-terminus suggests that it is comprised of four α-helices, tethered to the catalytic domain by a long unstructured linker (*Figure 6—figure supplement 1A*). The IQ motif (residues 127–136) resides in the third helix, separated from helix four by an unstructured loop. To determine if ORP3 might allosterically regulate IQSec1 activity, we mapped the ORP3 binding site on IQSec1 by mutagenesis (*Figure 6 - Figure 5—figure supplement 1A and B*). Note that all IQSec1 truncation mutants and IQ mutant tested co-precipitate ORP3. This analysis indicated that the ORP3 binding site is contained entirely within the unstructured loop (residues 152–212), as a mutant lacking this sequence fails to precipitate ORP3 (*Figure 6—figure supplement 1B*).

As described above, ORP3 contains an N-terminal PH domain, a C-terminal lipid transfer (ORD) domain, and two central FFAT motifs previously shown to mediate interaction with the ER-resident proteins VAP-A and VAP-B (*Figure 6—figure supplement 1C*). To determine how ORP3 binds IQSec1, we performed a co-precipitation analysis with a series of ORP3 mutants (*Figure 6—figure supplement 1C,D*). Interestingly, neither an N-terminal fragment comprising roughly the N-terminal half of ORP3 (1–445) nor an overlapping fragment comprising the C-terminal half (398-886) co-precipitated with IQSec1. However, extending the N-terminal construct to include the second FFAT motif (1-555) yielded a robust interaction with IQSec1 (*Figure 6—figure supplement 1D*), suggesting that the region encompassing this motif (161-454) mediates the

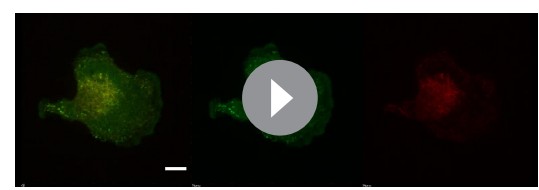

**Video 5.** MDA-MB-231 cell expressing GFP-ORP3 and mCherry-STIM1, imaged live for 30 min at a rate of 1 frame every 15 s. Thapsigargin was added 2.5 min after the start of the movie. Note the co-recruitment of ORP3 and STIM1 to the PM after thapsigargin treatment. These changes have been quantified in *Figure 4B*. Bar = 10 μm.
https://elifesciences.org/articles/54113#video5

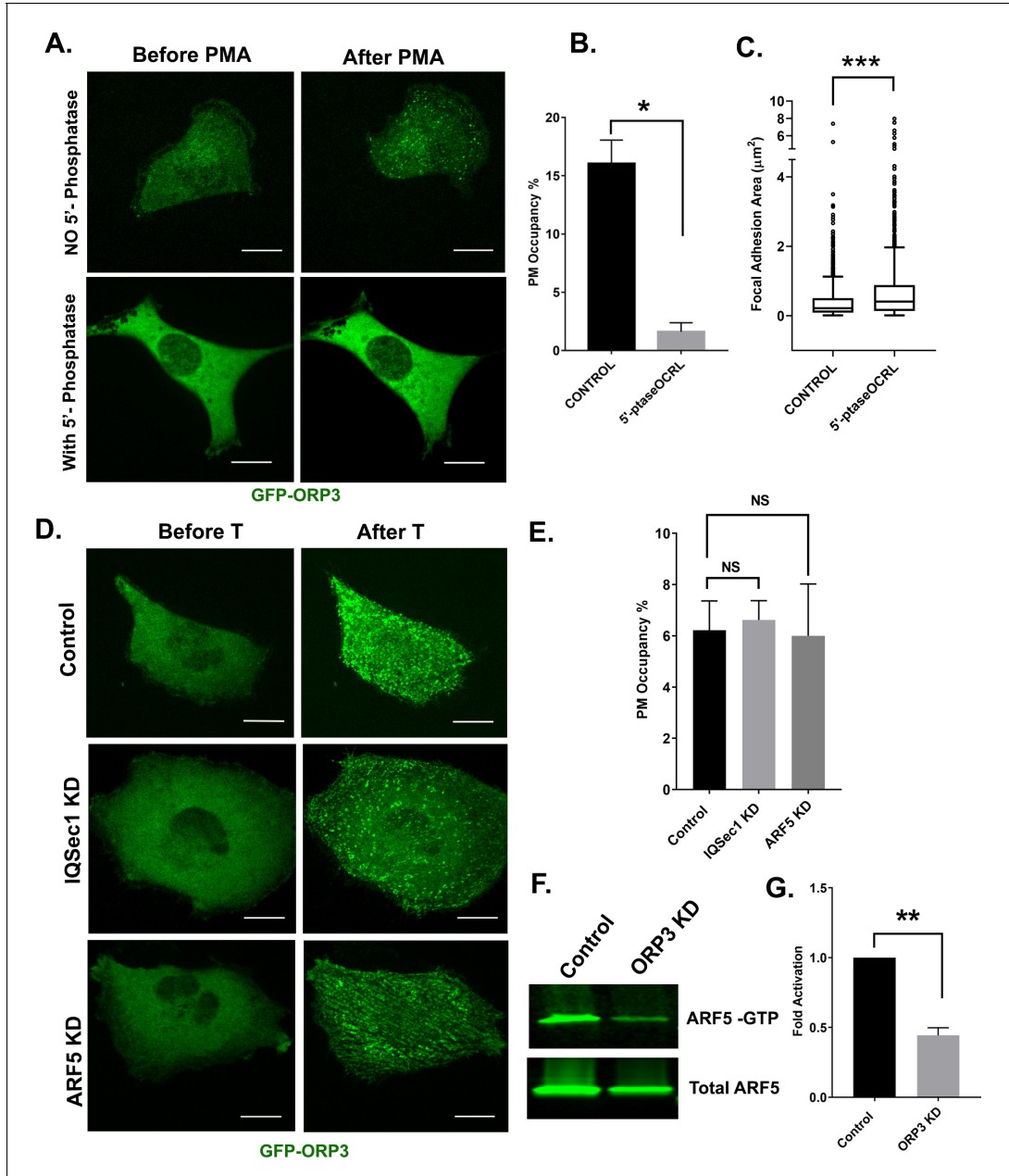

**Figure 5.** ORP3 requires PI(4,5)P$_2$ for recruitment to the PM, and acts upstream of IQSec1 to stimulate Arf5 activation. (A) Live cells expressing GFP-ORP3, mCherry-CRY2-5-Ptase$_{OCRL}$ and CINB-CAAX (two components of the blue light system) were stimulated with blue light to deplete PI(4,5)P$_2$ and then treated with PMA (2.5 μM) for 30 min. Images show individual cells before and after PMA treatment. Bar = 10 μm. (B) Fraction of PM area containing ORP3 puncta after optogenetic depletion of PI(4,5)P$_2$ and treatment with PMA. Controls were transfected with unfused mCherry and exposed to blue light similarly to cells expressing mCherry-CRY2-5-Ptase$_{OCRL}$. Data were collected from 12 cells per group, from three experiments. (C) Live cells expressing GFP-paxillin, mCherry-CRY2-5-Ptase$_{OCRL}$ and CINB-CAAX were stimulated with blue light to deplete PI(4,5)P$_2$ and focal adhesion size was measured before and after PI(4,5)P$_2$ depletion. Data were collected from 18 cells for each condition. See *Figure 4—figure supplement 1G* for representative images. (D) Control cells or cells depleted of either IQSec1 or Arf5 were transfected with GFP-ORP3, treated with thapsigargin (1 μM) and imaged live. Panels show cells before and after 5 mins of thapsigargin treatment. Bar = 10 μm. (E) Fraction of PM area containing GFP-ORP3 puncta after thapsigargin treatment corresponding to panel 'D'. Data were collected from 13 cells per group, from three experiments. (F) Representative blot of active Arf5 in lysates prepared from ORP3 depleted cells. (G) Quantitation of Arf5 activity compiled from three independent experiments.

The online version of this article includes the following figure supplement(s) for figure 5:

**Figure supplement 1.** Mapping the binding site for ORP3 on IQSec1.

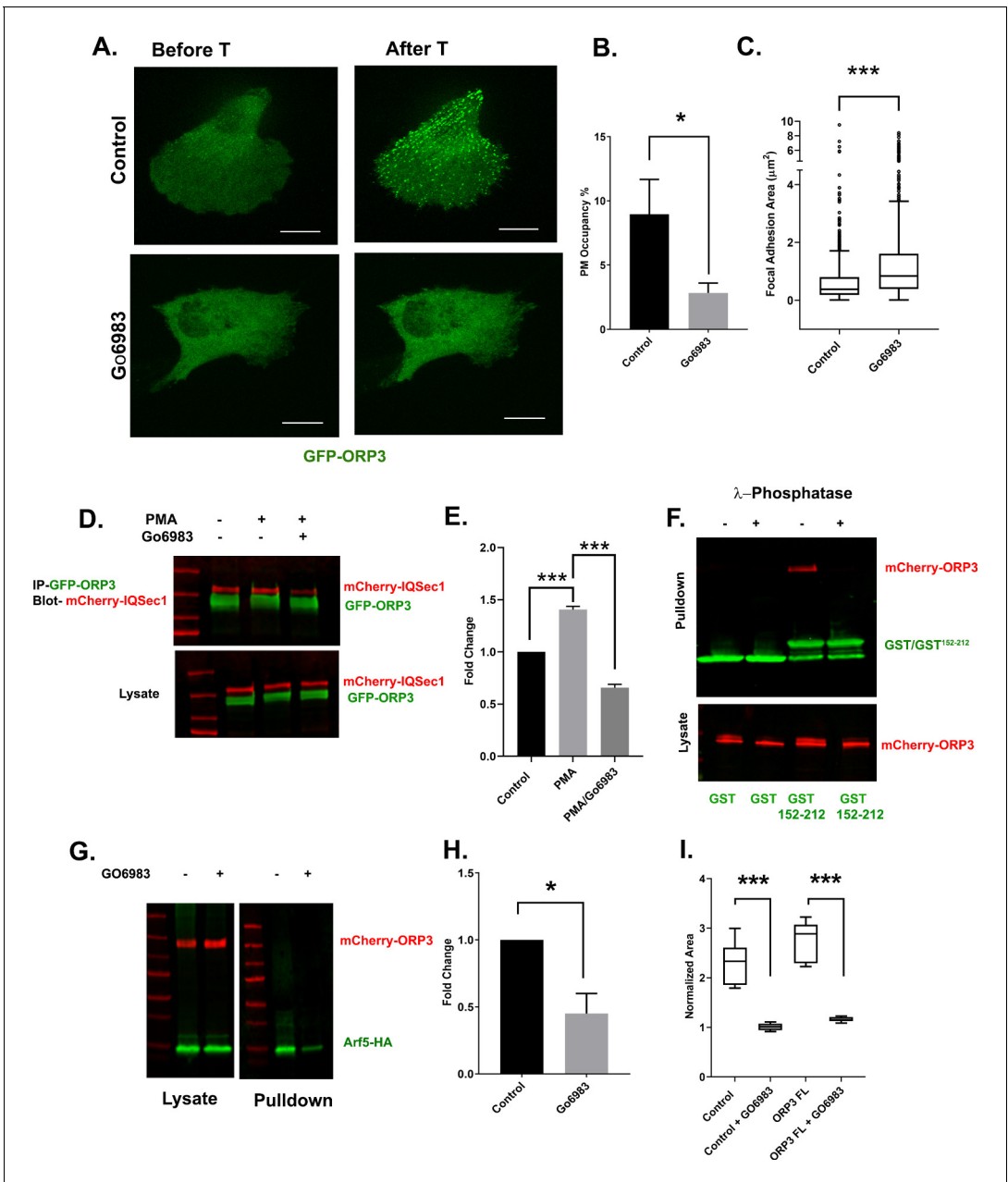

**Figure 6.** Phosphoregulation of ORP3 recruitment to the plasma membrane. (**A**) MDA-MB-231 cells expressing GFP-ORP3 were mock treated with DMSO or with the pan-PKC inhibitor Gö6983 (20 µM) for 2 hr. Both groups of cells were then stimulated with thapsigargin (1 µM) and imaged live. Images show a representative cell after 5 min thapsigargin treatment. Bar = 10 µm. (**B**) Fraction of the PM area containing ORP3 puncta from control and Gö6983 treated cells after thapsigargin treatment. Data were collected from 15 cells per group, from three experiments. (**C**) Quantification of focal adhesion size in control and Gö6983 treated cells, stained for endogenous vinculin as a marker for FAs. N for control = 848 and for Gö6983 = 1293 FAs. Data were collected from 36 cells per group, spanning three experiments. See *Figure 6—figure supplement 2* for representative images. (**D**) Cells co-expressing mCherry-IQSec1 and GFP-ORP3 were incubated with PMA in the presence or absence of Gö6983. ORP3 immunoprecipitates were then probed for IQSec1. Data from three independent experiments were quantified and presented in (**E**). (**F**) Lysates of cells expressing mCherry-ORP3 were incubated (or not) with λ-phosphatase to dephosphorylate ORP3, then precipitated with either GST alone or a GST fusion containing IQSec1 residues 152–212. This experiment was done twice and a similar trend was noted. For repeat see Figure S7B. (**G**) Arf5 activity was measured by pulldown in the presence and absence of the PKC inhibitor Gö6983. (**H**) Quantitation of Arf5 activity, corresponding to panel G. Data were compiled from three independent experiments. (**I**) Quantification of cell migration out of spheroids from parental MDA-MB-231 cells or cells stably expressing WT ORP3. One batch of spheroids from each group was left untreated, and a second batch was treated with Gö6983 after embedding in 3D collagen. Spheroids were imaged live in the continuous presence of the drug for 22 hr. Data were collected from seven spheroids from each group.

The online version of this article includes the following figure supplement(s) for figure 6:

*Figure 6 continued on next page*

**Figure supplement 1.** Mapping the interaction between ORP3 and IQSec1.
**Figure supplement 2.** Imaging of focal adhesions upon inhibition of PKC or PI4KIIIα.

interaction of ORP3 with IQSec1. Deletion of both FFAT motifs in full-length ORP3 resulted in reduced, but still detectable interaction with IQSec1 (*Figure 6—figure supplement 1D*), suggesting that these motifs may contribute to, but are not essential, for binding.

The involvement of FFAT motifs in binding of ORP3 to IQSec1 suggested the possibility that IQSec1 might interfere with the ability of ORP3 to bind VAP-A, thereby inhibiting formation of ER/PM contacts. Instead, we observed that VAP-A co-precipitates with ORP3 and IQSec1, and that increasing amounts of exogenous VAP-A actually *enhanced* co-precipitation of ORP3, suggesting that the three proteins form a complex in a cooperative manner (*Figure 6—figure supplement 1E–F*).

## Phosphoregulation of the ORP3/IQSec1 interaction

Despite the fact that ORP3 possesses both FFAT motifs that mediate interaction with the ER and a PH domain that is essential for interaction with the PM, the majority of ORP3 is diffusely cytosolic at steady state. This suggests that ORP3 may exist in an autoinhibited conformation in which both the FFAT motifs and PH domain are unavailable for interaction until released by an upstream stimulus. Previous studies have shown that both the calcium ionophore A23187 and the PKC activator phorbol myristate acetate (PMA) induce a phosphorylation-dependent mobility shift in ORP3 on SDS-PAGE gels, and that phosphorylation is important for the interaction of ORP3 with VAP-A (*Weber-Boyvat et al., 2015*).

Canonical PKC isoforms are activated by $Ca^{2+}$. To determine if PKC mediates translocation of ORP3 to ER/PM contacts in response to $Ca^{2+}$ influx, cells were pretreated with the broad-spectrum PKC inhibitor Gö6983 prior to thapsigargin treatment. As expected, in control cells thapsigargin stimulated rapid and robust recruitment of ORP3 to the PM (*Figure 6A–B*). In contrast, pretreatment with Gö6983 potently inhibited $Ca^{2+}$-stimulated recruitment, indicating that it is indeed dependent on PKC (*Figure 6A–B*). Interestingly, treatment with Gö6983 also increased focal adhesion size, phenocopying ORP3, IQSec1 and Arf5 knockdowns (*Figure 6C—figure supplement 2A*). Treating cells with PMA increased the phosphorylated pool of ORP3 (as indicated by a change in mobility of the green band, *Figure 6D*) and also stimulated the association of ORP3 with IQSec1. In contrast, incubation with Gö6983 reduced interaction of ORP3 with IQSec1 (*Figure 6D–E*).

Because these experiments were performed using lysates from cells that co-express ORP3 and IQSec1, it was formally possible that phosphorylation of ORP3, IQSec1, or both is required for interaction. To distinguish among these possibilities, lysates of cells expressing ORP3 were incubated with (or without) λ-phosphatase. The effectiveness of phosphatase treatment was validated by a noticeable shift in ORP3 mobility (*Figure 6—figure supplement 2B*). Lysates were then incubated with a GST fusion containing only the ORP3 binding domain of IQSec1 (residues 152–212). As shown in *Figure 6F*, no ORP3 was precipitated by GST alone under either condition, while it was clearly precipitated by $GST^{152-212}$ in the absence of phosphatase. In contrast, no detectable ORP3 was precipitated after λ-phosphatase treatment, strongly suggesting that this interaction requires phosphorylation of ORP3, but not IQSec1.

If ORP3 phosphorylation is required for its interaction with IQSec1, inhibition of PKC should attenuate Arf5 activation. As shown in *Figure 6G and H*, treatment of cells with Gö6983 did indeed inhibit Arf5 activity by ~50%. Again, this residual activity is likely due to activation of Arf5 by other GEFs that reside in the ERGIC or cis-Golgi.

We also reasoned that if PKC-mediated phosphorylation is required for ORP3 activity at the PM, treatment with PKC inhibitor should inhibit ORP3 dependent cell migration. In agreement with this hypothesis, cell migration out of MDA-MB-231 spheroids was completely inhibited by Gö6983 (*Figure 6I*). Together, these data indicate that phosphorylation of ORP3 by PKC mediates not only its PH domain-dependent recruitment to the PM, but also the co-operative association of ORP3/ VAPA/IQSec1 specifically at ER/PM contact sites (*Figure 6—figure supplement 1E–F*) and the downstream activation of Arf5, which are all required for cell migration.

## ORP3 extracts PI4P from the plasma membrane

All other members of the ORP family that have been characterized mediate the exchange of lipids at membrane contact sites between the ER and either another organelle or the plasma membrane. All ORPs (except ORP2) appear to share the ability to transfer PI4P from the target organelle to the ER, where the Sac1 phosphatase converts PI4P to phosphatidylinositol, but differ in the cargo that is exchanged for PI4P; OSBP and ORP2 transfers sterols (*Mesmin et al., 2013*; *Wang et al., 2019*), while ORP5, ORP8 and ORP10 preferentially transfer PS (*Chung et al., 2015*; *Venditti et al., 2019*). However, the lipid transfer properties of ORP3 have not as yet been characterized. If ORP3 acts as a lipid transfer protein at the plasma membrane, it would be predicted to extract PI4P at this site. To determine if this is the case, we used the selective PI4P probe GFP-P4M2X to detect endogenous PI4P in live cells (*Hammond et al., 2014*). Cells were co-transfected with GFP-P4M2X and either unfused mCherry or mCherry-ORP3, then treated with thapsigargin to stimulate translocation of ORP3 to the PM. Cells were imaged by spinning disc confocal microscopy at a single focal plane containing the ventral plasma membrane. As previously reported, PI4P is abundant at the plasma membrane at steady state (*Figure 7A*). In control cells expressing unfused mCherry, thapsigargin treatment did not substantially alter the level of plasma membrane-associated GFP-P4M2X (*Figure 7A–B* and *Video 6*). In contrast, thapsigargin induced rapid translocation of mCherry-ORP3 to plasma membrane puncta and a corresponding dramatic decrease in plasma membrane PI4P (*Figure 7C–D* and *Video 7*). Importantly, a truncated ORP3 mutant lacking the ORD domain (mCherry-ORP3$^{1-555}$) was robustly recruited to the PM in response to thapsigargin but did not extract PI4P (*Figure 7E–F*). This observation also enabled us to rule out any complication due to competition between the ORP3 PH domain and the P4M2X probe.

To determine if ORP3-mediated lipid transfer is important for FA disassembly we treated cells with GSK-A1, which selectively depletes the PM pool of PI4P and would thus inhibit ORP3 mediated lipid exchange at ER-PM contact sites. As shown in *Figure 7G–H* and *Figure 6—figure supplement 2A*, GSK-A1 treatment not only increased focal adhesion size in 2D culture but also inhibited the migration of MDA-MB-231 cells from tumor spheroids.

## ORP3 binds phosphatidylcholine

As noted above, all ORPs exchange a phosphoinositide extracted from the target membrane for another lipid that is delivered from the ER to the target. To identify the lipid cargo(s) that is exchanged for PI4P at the PM, we performed lipid mass spectroscopy. For this purpose, cells were transfected with either GFP alone or GFP-ORP3, then treated (or not) with thapsigargin to drive ORP3 to the PM. Cells were then lysed and incubated with beads containing anti-GFP nanobodies. After washing, immunoprecipitates were extracted with methanol:choloroform (2:1) mixture, and the extracted lipids resolved by mass spectroscopy.

Structural modeling has suggested that the ORP3 lipid binding pocket is too small to accommodate sterols and does not fit the consensus for binding to phosphatidylserine (*Chung et al., 2015*; *Perry and Ridgway, 2006*; *Vihervaara et al., 2011*; *Wang et al., 2019*). In agreement with these predictions, ORP3 did not detectably bind sterols (not shown) and only weakly bound PS, PE or PG (*Figure 7I* and *Figure 7—source data 1*). In no case was binding enhanced following treatment with thapsigargin. Similarly, no significant interactions were observed with sphingomyelins (SM). In contrast, several species of phosphatidylcholine were detected in ORP3 precipitates, in amounts that were significantly increased in the thapsigargin-treated samples. The predominant PC species recovered were 16:0/18:1 and 16:0/16:0. Importantly, the levels of both were increased 3-fold upon treatment with thapsigargin, indicating that calcium influx stimulates cargo loading. Together, these observations suggest that ORP3 exchanges PI4P for PC at ER/PM contact sites, in a calcium-dependent manner (*Figure 7I*).

## Functions of ORP3 in focal adhesion turnover

To determine which aspects of ORP3 function are important for adhesion turnover we performed rescue experiments with ORP3 mutants in ORP3-depleted cells. As shown in *Figure 7J*, the decrease in FA disassembly rate observed in ORP3 deficient cells was completely rescued by re-expressing wild-type ORP3. In contrast, a mutant lacking both FFAT motifs ($\Delta^{FFAT}$) failed to restore adhesion turnover, confirming that interaction of ORP3 with VAP-A/B (and the formation of ER/PM contacts)

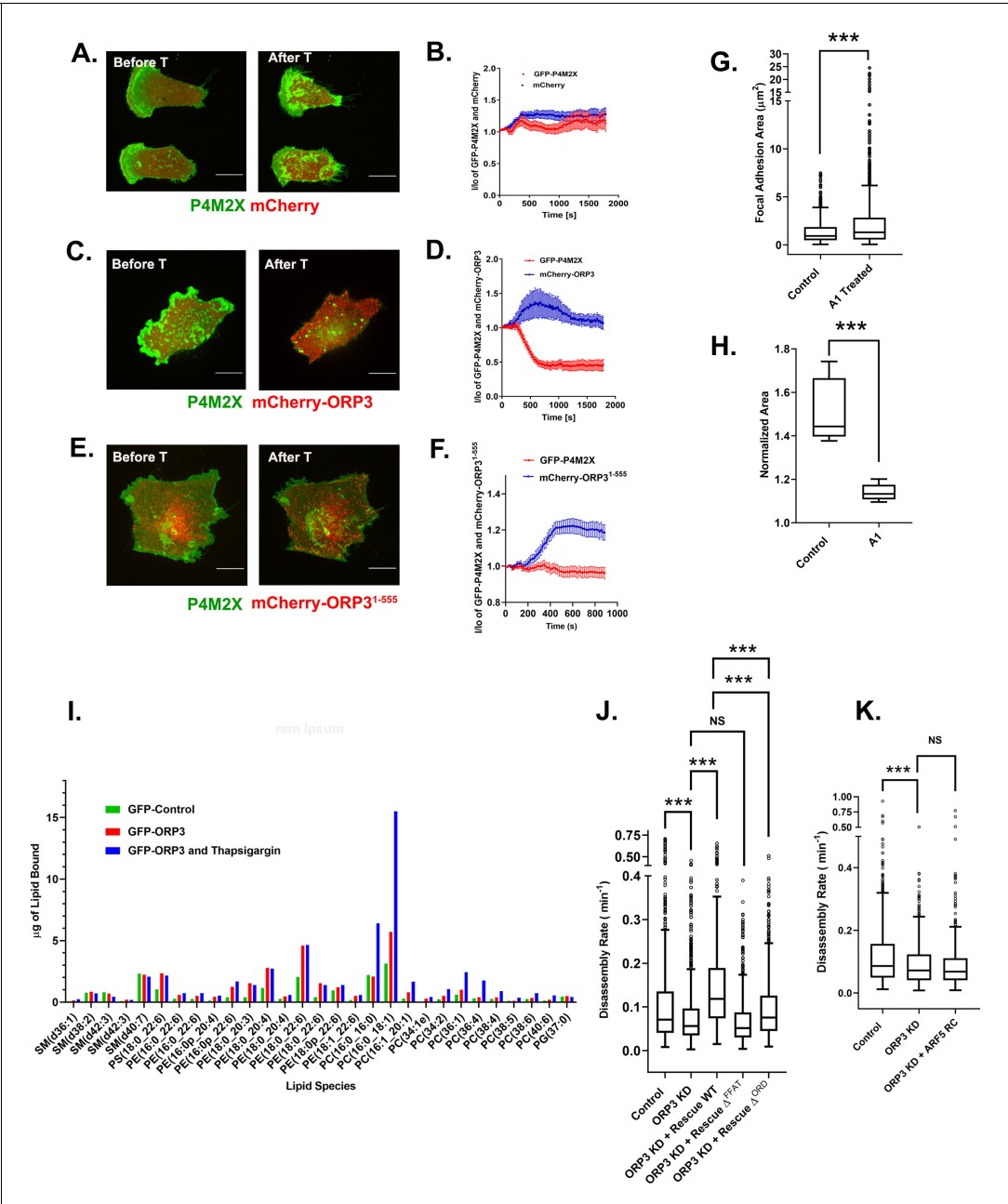

**Figure 7.** ORP3 extracts PI4P from the plasma membrane in exchange for PC. (**A–F**) MDA-MB-231 cells expressing the PI4P probe GFP-P4M2X and either mCherry alone (**A**) mCherry-ORP3 (**C**) or mCherry-ORP3$^{1-555}$ (**E**) were imaged live. Images show stills before (Frame 1) and after (Frame 75) treatment with thapsigargin (1 µM) for 30 mins. The changes in PM fluorescence for GFP-P4M2X (red) or mCherry (blue) were quantified and presented in (**B**), (**D**) and (**F**) respectively. Data were compiled from 20 regions of interest from three independent experiments. (**G**) Cells were treated with GSK-A1 to deplete PM PI4P and focal adhesion size was measured in control (695 FAs) and GSK-A1 treated cells (1019 FAs). Data were collected from 30 cells per group, spanning three experiments. For corresponding images refer to Figure S7A. (**H**) Quantification of cell migration out of spheroids from control and GSK-A1-treated cells after 22 hr in 3D collagen gels. Data were collected from six spheroids for each group. (**I**) Quantification of lipids precipitated with unfused GFP (green) or GFP-ORP3 before (red) and after (blue) thapsigargin treatment. (**J**) FA disassembly rates in cells expressing GFP-paxillin. ORP3 depleted cells were transfected with constructs encoding WT ORP3 or mutants lacking the FFAT motifs ($\Delta^{FFAT}$) or the ORD domain ($\Delta^{ORD}$). FA turnover was measured as described in *Figure 1*. N for control = 828, ORP3 KD = 1504, ORP3 KD+ Rescue WT = 677, ORP3 KD+ Rescue $\Delta^{FFAT}$ = 792, ORP3 KD+ Rescue $\Delta^{ORD}$ = 657 focal adhesions. Data were compiled from 15 cells per group spanning three independent experiments. Note that the disassembly rates for ORP3 KD+ Rescue WT are significantly higher than ORP3 KD and reach control level compared to WT ORP3 KD+ Rescue $\Delta^{ORD}$ that significantly rescues over ORP3 KD but is not as efficient as WT ORP3. (**K**) FA disassembly rates were measured in ORP3 depleted

*Figure 7 continued on next page*

*Figure 7 continued*

cells expressing the rapid cycling T161A mutant of Arf5. N for control = 564, ORP3 KD = 551, ORP3 KD + ARF5 RC = 467. Data were compiled from 15 cells per group spanning three independent experiments.

The online version of this article includes the following source data and figure supplement(s) for figure 7:

**Source data 1.** Mass spectroscopy of lipids bound to ORP3 – data underlying *Figure 7I*.
**Figure supplement 1.** Model summarizing the finding of this study.

is essential for its function. Interestingly, reconstitution with an ORP3 mutant lacking the ORD domain partially rescued the defect, but was clearly not as effective as the wild-type protein (*Figure 7J*). This result implies that the lipid transfer function of ORP3 enhances, but is not essential for FA disassembly.

Based on this observation, we reasoned that ORP3 might serve two independent functions, the first being the localization and allosteric activation of IQSec1, the second being localized lipid exchange in proximity to FAs. To test this hypothesis, we depleted cells of ORP3 and performed rescue experiments with rapid cycling Arf5, which as described above can restore adhesion disassembly in IQSec1-depleted cells. Interestingly, rapid cycling Arf5 did not rescue the FA disassembly defects observed in ORP3 depleted cells (*Figure 7K*). Together these data suggest that the IQSec1-mediated activation of Arf5 is not sufficient for FA disassembly and also depends on ORP3 mediated lipid exchange at ER/PM contact sites.

## Discussion

Arf family GTPases have long been implicated in the regulation of cell migration. It is well accepted that Arf6 mediates the recycling of β1-integrins from endosomal compartments (*Oh and Santy, 2010*; *Powelka et al., 2004*), and Arf1 has been reported to promote focal adhesion assembly (*Norman et al., 1998*), but whether and how Arfs may control focal adhesion disassembly remains poorly understood. Interestingly, at least six Arf GAPs (GIT1, GIT2, ASAP1, ASAP2, ASAP3, ARAP2, ACAP2) have been localized to focal adhesions, suggesting that maintaining a low local level of Arf activity may be important for adhesion stability (*Hoefen and Berk, 2006*; *Liu et al., 2006*; *Randazzo et al., 2000*; *Ha et al., 2008*; *Vitali et al., 2019*; *Zhao et al., 2000*).

Here we report that activation of Arf5 by the Arf GEF IQSec1 is crucial for FA disassembly. Depletion of either IQSec1 or Arf5 significantly reduces the rate of adhesion turnover, resulting in the formation of large, stable adhesions and near cessation of cell motility. Despite the high level of sequence similarity among the Arfs (65% identity), expression of rapid cycling Arf5, but not Arf1 or Arf6, largely restores adhesion disassembly in IQSec1-deficient cells. This was unexpected, as Arf5 is generally thought to function in the early secretory pathway, where it is activated by a different Arf GEF, GBF1 (*Chun et al., 2008*; *Claude et al., 1999*; *Volpicelli-Daley et al., 2005*). However, our finding that ER/PM contact sites cluster near focal adhesions supports a novel role for Arf5 at the PM.

### Regulation of IQSec1 by calcium

The IQSec subfamily of Arf GEFs are characterized by the presence of an N-terminal IQ motif that binds calmodulin in a non-canonical calcium

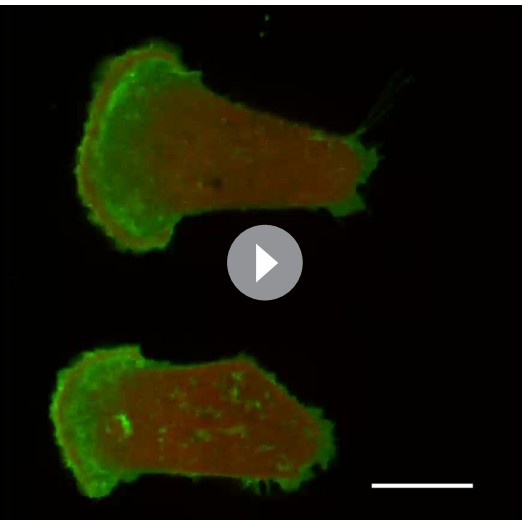

**Video 6.** MDA-MB-231 cell expressing GFP-P4M2X (probe for phosphatidylinositol-4-phosphate/PI4P) and unfused mCherry was imaged live for 30 min at a rate of 1 frame every 15 s. Thapsigargin was added 2.5 min after the start of the movie. Bar = 10 μm. Note that PM PI4P levels remain unchanged after addition of thapsigargin (*quantified in Figure 7B*).
https://elifesciences.org/articles/54113#video6

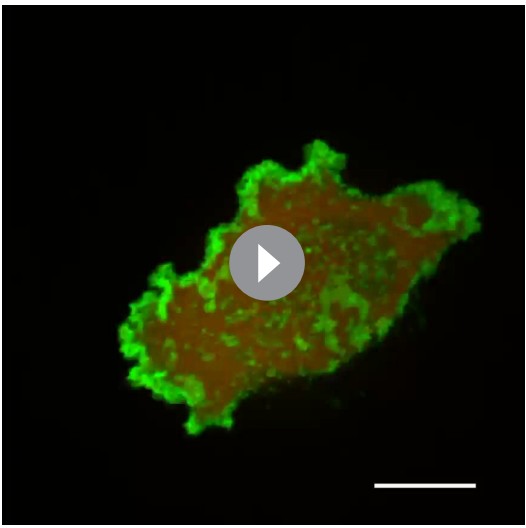

**Video 7.** MDA-MB-231 cell expressing GFP-P4M2X and mCherry-ORP3 was imaged live for 30 min at a rate of 1 frame every 15 s. Thapsigargin was added 2.5 min after the start of the movie. Notice the loss of GFP-P4M2X from the PM with the concomitant recruitment of mCherry-ORP3 (*quantified in Figure 7D*). The reduction in overall probe intensity is presumably due to transfer of PI4P to the ER, where it is dephosphorylated by Sac1. The probe is redistributed from the PM to what are presumably endosomes. Bar = 10 µm.

https://elifesciences.org/articles/54113#video7

independent manner (*Myers et al., 2012*). In the case of neuronal IQSec2, influx of calcium via NMDA receptors triggers the release of calmodulin and stimulates IQSec2 GEF activity. Since the sequence flanking the IQ motif is nearly identical in all three IQSec isoforms, we hypothesized that calcium would also enhance the activity of the ubiquitously expressed IQSec1 in nonneuronal cells. Here we show that store-operated calcium influx via STIM1/Orai1 channels significantly stimulates IQSec1 activity, and that basal activity is inhibited in the presence of the selective Orai1 channel inhibitor BTP2.

SOCE also serves to recruit the IQSec1 binding partner, ORP3 to the PM. Intriguingly, the association of ORP3 with IQSec1 occurs through an unstructured loop adjacent to the IQ motif, which is both necessary and sufficient for ORP3 binding. This suggests a two-step activation mechanism for IQSec1 in which calmodulin displacement during $Ca^{2+}$ influx may be required for ORP3 binding and full activation. Future studies will be required to define this mechanism in more detail.

## Phosphoregulation of ORP3 recruitment to the plasma membrane

Although it possesses both a PH domain that mediates interaction with $PIP_2$ at the PM and two FFAT motifs that mediate interaction with VAP-A at the ER, ORP3 is primarily cytosolic at steady state, especially if calcium influx is inhibited using BTP2. Previous studies have shown that ORP3 becomes hyperphosphorylated upon treatment of cells with PMA, and that this significantly enhances its interaction with VAP-A (*Weber-Boyvat et al., 2015*). Here we show that cytosolic ORP3 rapidly translocates to ER/PM contacts in a PKC-dependent manner. Conversely, pharmacological inhibition of PKC renders ORP3 diffusely cytosolic even in the presence of thapsigargin, and increases its mobility on SDS-PAGE gels. Although we cannot rule out that PKC triggers ORP3 phosphorylation indirectly (e.g. as part of a kinase cascade), bioinformatics analysis (NetPhos 3.0) predicts ~20 PKC sites scattered throughout the ORP3 sequence. Previously, Olkonnen and colleagues (*Weber-Boyvat et al., 2015*) identified a number of phosphorylated serines by mass spectroscopy, only one of which (S372) fits the consensus for a PKC site. These authors reported that none of the identified sites altered mobility of ORP3 on SDS-PAGE gels when mutated to alanine, but they did not determine whether interaction with VAP-A was affected.

Together, these observations suggest that cytosolic ORP3 exists in a soluble autoinhibited state in which its PH domain and FFAT motifs are sequestered. We hypothesize that PKC-mediated phosphorylation induces a conformational change that exposes these domains simultaneously, thereby providing a mechanism for the regulated formation of ER/PM contacts in response to calcium influx. Interestingly, phosphoregulation of lipid transfer activity has also been reported for OSBP and the ceramide transporter CERT, although in both cases phosphorylation actually inhibits transfer (*Goto et al., 2012*; *Kumagai et al., 2007*). Studies are in progress to map ORP3 phosphorylation sites and determine their importance in the recruitment of the ORP3/IQSec1 complex to the plasma membrane.

IQSec1 also preferentially associates with the hyperphosphorylated form of ORP3, implying that ORP3, VAP-A/B and IQSec1 form a tripartite complex upon ORP3 phosphorylation. This hypothesis is supported by our observation that VAP-A co-precipitates with ORP3 and IQSec1, and that increased expression of VAP-A actually enhances interaction of IQSec1 with ORP3. Conversely,

mutation of the FFAT motifs in ORP3 reduces binding of IQSec1 to ORP3, suggesting that either VAP-A interaction is a prerequisite for IQSec1 binding, or that the two proteins bind cooperatively to phosphorylated ORP3. Importantly, inhibition of PKC not only reduces interaction between IQSec1 and ORP3, but also significantly attenuates Arf5 activity, further supporting the notion that ORP3 (and possibly VAP-A) binding positively regulates IQSec1 activity.

## Lipid transfer by ORP3

All 12 mammalian ORPs contain a C-terminal ORD (OSBP-related) domain that is thought to mediate the exchange of lipids between closely apposed membranes. A unifying feature of the ORDs is the presence of a signature sequence for the binding of PI4P, and in most cases that have been examined PI4P is indeed one of the cargos. In these cases, PI4P is transferred from the target membrane to the ER, where it is dephosphorylated by the resident protein Sac1 (*Nemoto et al., 2000*; *Zewe et al., 2018*). In the case of OSBP, PI4P is exchanged for cholesterol, which is transferred to the TGN or endosomes from the ER (*Mesmin et al., 2013*; *Sobajima et al., 2018*). ORP5 and ORP8 have been shown to transfer phosphatidylserine to the PM in exchange for PI4P (*Chung et al., 2015*), while ORP10 transfers PS from the ER to the TGN (*Venditti et al., 2019*). Recently, ORP2 was reported to transfer cholesterol to the PM in exchange for $PI(4,5)P_2$ rather than PI4P (*Wang et al., 2019*).

Here, we report that ORP3 extracts PI4P from the PM in a calcium-dependent manner. Thapsigargin treatment triggers a dramatic decrease in the level of PM PI4P that corresponds temporally with delivery of ORP3 to ER/PM contact sites. Surprisingly, mass spectroscopy of lipids bound to ORP3 determined that the primary cargo is PC, strongly suggesting that ORP3 exchanges PC for PI4P. However, unlike other lipid species for which fluorescent probes exist, such probes are not available for endogenous PC, and it has not been possible for us to track the movement of PC from the ER to the PM in intact cells. Moreover, most other reports describing the properties of lipid transfer proteins have used in vitro assays that measure the transfer of a lipid species from one population of liposomes to another, again using fluorescent probes (*Chung et al., 2015*; *Mesmin et al., 2013*). Such liposomes, which mimic biological membranes, invariably contain a large fraction of PC, rendering in vitro analysis of PC transfer problematic.

It is important to note, however, that pharmacological depletion of PI4P from the PM using GSK-A1 actually resulted in larger adhesions and reduced motility in the spheroid assay. This suggests that it is not extraction of PI4P, but rather local insertion of PC into the PM that is important for adhesion disassembly. The most abundant PC species identified in our mass spectroscopy analysis were POPC (16:0/18:1) and DPPC (16:0/16:0), which are among the most abundant PC species in eukaryotic cells (*Harayama and Riezman, 2018*). How might insertion of PC contribute to focal adhesion turnover? The lipid environment at FAs is reported to be highly ordered (*Gaus et al., 2006*), at least in part due to the accumulation of cholesterol and negatively charged lipids in ordered domains around clustered integrins (*Kalli et al., 2017*). One possibility is that insertion of a high concentration of PC into this environment, particularly in the cytoplasmic leaflet, would dilute the concentration of cholesterol or charged lipids in adhesions, rendering the bilayer more fluid. Alternatively, many focal adhesion components (talin, vinculin, FAK) bind to acidic lipids (e.g. $PIP_2$), and dilution of negatively charged headgroups could reduce their respective affinities for the membrane at these sites. Future work will be directed at dissecting these potential mechanisms.

## Integration of ORP3, IQSec1 and Arf5 in focal adhesion turnover

The proximity of ORP3-containing contact sites to focal adhesions, and our observation that ORP3 knockdown slows adhesion disassembly, suggest that local lipid transfer is essential for efficient adhesion turnover. Surprisingly, we found that an ORP3 mutant lacking the ORD domain partially restored adhesion disassembly in cells depleted of endogenous ORP3. This observation suggests that, while lipid transfer is indeed important, it is not the only role of ORP3 in this process. In contrast, a mutant lacking both FFAT motifs completely failed to restore adhesion turnover, indicating that formation of ER/PM contacts is critical. Interestingly, this mutant also has a substantially reduced affinity for IQSec1, suggesting that both lipid transfer and activation of Arf5 at contact sites are essential for adhesion disassembly. In support of this hypothesis, expression of rapid cycling Arf5,

which did restore adhesion turnover in IQSec1-depleted cells, was not sufficient in ORP3-deficient cells.

Based on these observations, we propose a model (*Figure 7—figure supplement 1*) in which local calcium influx triggers the PKC-dependent assembly of an ORP3/VAP-A/IQSec1 complex and its translocation to ER/PM contact sites containing STIM1/Orai1 calcium channels that are in close proximity to focal adhesions. Lipid transfer mediated by ORP3, combined with Arf5 activation by IQSec1, drives the downstream disassembly of nearby adhesions and the subsequent ability of cells to detach their trailing edge, allowing migration to proceed.

An important question that remains unresolved is the role of Arf5 in FA disassembly. Specifically, what are the downstream effectors of Arf5 at this location and how do they promote adhesion turn-over? One previous study determined that the PIP5 kinase isoform PI(4)P 5-kinaseIIIβ, which is allosterically activated by Arfs (*Honda et al., 1999*) drives the recruitment of the endocytic machinery at sites of adhesion disassembly through local production of PI(4,5)P$_2$ (*Chao et al., 2010*). Determining whether Arf5 stimulates local PI(4,5)P$_2$ production via PI(4)P 5-kinaseIIIβ in intact cells, or whether other unidentified Arf5 effectors drive adhesion disassembly will be an important goal for future research.

# Materials and methods

**Key resources table**

| Reagent type (species) or resource | Designation | Source or reference | Identifiers | Additional information |
|---|---|---|---|---|
| Cell line (*H. sapiens*) | MDA-MB-231 | ATCC | CRM-HTB-26 | Human breast cancer cell line |
| Cell line (*H. sapiens*) | HEK293 | ATCC | CRL-1573 | Human embryonic kidney cell line |
| Cell line (*H. sapiens*) | HT1080 | ATCC | CCL-121 | Human fibrosarcoma cell line |
| Transfected construct (*H. sapiens*) | FLAG-IQSec1 (NP_001127854.1) | Genscript, Piscataway, NJ | | Plasmid to transiently express IQSec1 |
| Transfected construct (*H. sapiens*) | mCherry-ORP3 | Vesa Olkonnen **Lehto et al., 2005** | | Plasmid to transiently express mCherry-ORP3 |
| Transfected construct (*H. sapiens*) | Xpress tagged FFAT double mutant of ORP3 | Vesa Olkonnen **Lehto et al., 2005** | | Plasmid to transiently express ORP3ΔFFAT |
| Transfected construct (*H. sapiens*) | Xpress tagged PH domain mutant of ORP3 | Vesa Olkonnen **Lehto et al., 2005** | | Plasmid to transiently express ORP3ΔPH |
| Transfected construct (*H. sapiens*) | Xpress tagged ORP3$^{1-445}$ | Vesa Olkonnen **Lehto et al., 2005** | | Plasmid to transiently express ORP3 truncation mutant 1–445 |
| Transfected construct (*H. sapiens*) | Xpress tagged ORP3$^{398-886}$ | Vesa Olkonnen **Lehto et al., 2005** | | Plasmid to transiently express ORP3 truncation mutant 398–886 |
| Transfected construct (*H. sapiens*) | Xpress tagged ORP3$^{1-555}$ | Vesa Olkonnen **Lehto et al., 2005** | | Plasmid to transiently express ORP3 truncation mutant 1–555 |
| Transfected construct (*H. sapiens*) | GFP-Paxillin | Rick Horwitz (Allen Research Institute, Seattle, WA) | | Plasmid to transiently express GFP-paxillin |
| Transfected construct (*H. sapiens*) | DsRed-Paxillin | Rick Horwitz (Allen Research Institute, Seattle, WA) | | Plasmid to transiently express DsRed-paxillin |
| Transfected construct (*H. sapiens*) | mCherry-STIM1 and | Isabelle Derre (University of Virginia) | | Plasmid to transiently express mCherry-STIM1 |
| Transfected construct (*H. sapiens*) | YFP-VAP-A | Isabelle Derre (University of Virginia) | | Plasmid to transiently express YFP-VAP-A |

*Continued on next page*

*Continued*

| Reagent type (species) or resource | Designation | Source or reference | Identifiers | Additional information |
|---|---|---|---|---|
| Transfected construct (*H. sapiens*) | mCherry-CRY2-5-PtaseOCRL and | Olaf Idevall-Hagren (Uppsala University, Sweden) **Idevall-Hagren et al., 2012** | | Plasmid for optogenetic activation of OCRL |
| Transfected construct (*H. sapiens*) | CINB-CAAX | Olaf Idevall-Hagren **Idevall-Hagren et al., 2012** | | Plasmid for optogenetic activation of OCRL |
| Transfected construct (*L. pneumophila*) | GFP-P4M2X | Tamas Balla (NIH, Bethesda, Maryland) **Hammond et al., 2014** | | Biosensor for PI4P |
| Transfected construct (*H. sapiens*) | GFP- E-Syt1 | Pietro De Camilli (Yale Univ. New Haven, CT) | | Plasmid to transiently express GFP-E-Syt1 |
| Transfected construct (*H. sapiens*) | GFP-PLCδ | David Castle (Univ. Virginia) | | Biosensor for PI(4,5)P$_2$ |
| Antibody | mouse monoclonal antibody to HA (16B12) | Biolegend (San Diego, CA) | | WB (1:1000) |
| Antibody | rabbit polyclonal to GFP | ThermoFisher Scientific (Rockford, Il) | (A10262) | WB (1:2000) |
| Antibody | Mouse monoclonal anti-Xpress | ThermoFisher Scientific (Rockford, Il) | (R910-25) | WB (1:1000) |
| Antibody | Rabbit polyclonal antibody to mCherry | Biovision (San Francisco, CA) | (5993) | WB (1:1000) |
| Antibody | Mouse monoclonal antibody to tubulin | Sigma-Aldrich | (T9026) | WB(1:10,000) |
| Antibody | mouse monoclonal antibody to vinculin | Sigma-Aldrich | (V9131) | IIF (1:400) |
| Antibody | Mouse monoclonal antibody to human β1-integrin | BD Biosciences (Mississauga, ON, Canada) | (610467) | IIF (1:250) |
| Antibody | mouse antibody to GFP | Santa Cruz | (sc-9996) | WB (1:1000) |
| Antibody | mouse antibody to ORP3 | Santa Cruz | (sc-514097) | WB (1:1000) |
| Antibody | rabbit antibody to FAK | Santa Cruz | (sc-557) | IIF (1:100) |
| Antibody | Alexa Fluor 488 donkey anti-mouse secondary antibody | ThermoFisher Scientific (Rockford, Il) | (A10037) | IIF (1:500) |
| Antibody | Alexa Fluor 568 donkey anti-mouse secondary antibody | ThermoFisher Scientific (Rockford, Il) | (A21202) | IIF (1:500) |
| Antibody | Alexa Fluor 488 donkey anti-rabbit secondary antibody | ThermoFisher Scientific (Rockford, Il) | (A10042) | IIF (1:500) |
| Antibody | Alexa Fluor 568 donkey anti-rabbit secondary antibody | ThermoFisher Scientific (Rockford, Il) | (A21206) | IIF (1:500) |
| Antibody | IRDye 800CW Secondary Antibody | Li-COR | | WB (1:10,000) |
| Antibody | IRDye 650 Secondary Antibody | Li-COR | | WB (1:10,000) |
| Chemical compound, drug | Alexa Flour 647 phalloidin | ThermoFisher Scientific (Rockford, Il) | (A22287) | Dilution of 1:1000 |
| Chemical compound, drug | Thapsigargin | Tocris (Bristol, UK) | (1138) | 1 µM |
| Chemical compound, drug | Phorbol 12-myristate 13-acetate (PMA) | Tocris (Bristol, UK) | (1201/1) | 2.5 µM |

*Continued on next page*

*Continued*

| Reagent type (species) or resource | Designation | Source or reference | Identifiers | Additional information |
|---|---|---|---|---|
| Chemical compound, drug | YM 58483 (BTP2) | Tocris (Bristol, UK) | (3939/10) | 25 µM |
| Chemical compound, drug | Gö6983 | Abcam (Cambridge, UK) | (ab144414) | 20 µM |
| Commercial assay or kit | Odyssey PBS Blocking Buffer | Li-COR | | For immunoblotting 1:1 diluted in PBST (0.1% tween-20) |
| Commercial assay or kit | Q5 Site-Directed Mutagenesis Kit | NEB | | |
| Software, algorithm | NEBaseChanger | NEB | | Software to design primers for site-directed mutagenesis |
| Other | GFP-TRAP_A beads | Chromotek (Germany) | gta-10 | For precipitation of GFP-tagged proteins (25 µl of beads) |
| Other | Protein A Beads | (Cell Signalling, Danvers, MA) | (9863) | For immunoprecipitation (25 µl of bead) |

## Cells and antibodies

MDA-MB-231 cells, HT-1080 cells and HEK-293 cells obtained from the ATCC were cultured at 37°C with 5% $CO_2$ in high glucose DMEM supplemented with 10% FBS and antibiotics/antimycotics. The following antibodies were used in the study: mouse monoclonal antibody to HA (16B12) from Biolegend (San Diego, CA); rabbit polyclonal to GFP (A10262) and mouse monoclonal antibody to Xpress tag (R910-25) were from ThermoFisher Scientific (Rockford, Il); Rabbit polyclonal antibody to mCherry (5993) was from Biovision (San Francisco, CA); Mouse monoclonal antibody to tubulin (T9026), rabbit polyclonal antibody to IQSec1 (G4798) and mouse monoclonal antibody to vinculin (V9131) were from Sigma. Mouse monoclonal antibody to β1-Integrin (610467) was from BD Biosciences (Mississauga, ON, Canada), mouse antibody to GFP(sc-9996), mouse antibody to ORP3 (sc-514097) and rabbit antibody to FAK (sc-557) were from SantaCruz Biotechnologies (Dallas, Texas). Alexa Fluor 488 or 568 donkey anti-mouse or donkey anti-rabbit secondary antibodies (A10037, A21202, A10042 and A21206) were purchased from ThermoFisher Scientific (Rockford,Il).

## Chemicals and inhibitors

Thapsigargin, Phorbol 12-myristate 13-acetate (PMA) and YM 58483 (BTP2) were purchased from Tocris (Bristol, UK). Gö6983 was purchased from Abcam (Cambridge, UK). GSK-A1 was purchased from SYNkinase (Parkville, Australia). Alexa flour 647 Phalloidin (A22287) was purchased from ThermoFisher Scientific (Rockford, Il).

## Transfections

HEK-293 cells were transfected using Polyjet reagent from SignaGen (Montgomery, MD). MDA-MB-231 cells were transfected either using Lipofectamine 2000 (ThermoFisher Scientific (Rockford, Il) or using nucleofection reagent from Lonza. Briefly, $1 \times 10^6$ cells were mixed with 1–5 µg of the indicated expression vector and pulsed using the A-023 program on an Amaxa nucleofector system. Cells were used 24–48 hr after transfection.

## Plasmids

FLAG-IQSec1 (NP_001127854.1) was purchased from Genscript (Piscataway, NJ). GFP-IQSec1 was prepared by sub-cloning from Flag-IQSec1 into HindIII and EcoRI sites of pEGFP-C1. mCherry-IQSec1 was prepared by sub-cloning from Flag-IQSec1 in to the HindIII and EcoRI sites of pmCherry-C1. pmCherry-ORP3, Xpress tagged FFAT double mutant of ORP3, PH mutant of ORP3 and truncation mutants of ORP3 (1–445, 398–886, 1–555) were gifts from Vesa Olkonnen (University of Helsinki, Finland) and the generation of these reagents has been described in *Lehto et al. (2005)*. GFP-ORP3 was prepared by sub-cloning from pmCherry-ORP3 into the EcoRI site of pEGFP-C2. GFP-ORP3Δ^FFAT and GFP-ORP3Δ^ORD (1-555) were prepared by sub-cloning into the EcoRI and KpnI

sites of pEGFP-C2. GFP-paxillin and DsRed-paxillin were gifts from Rick Horwitz (Allen Research Institute, Seattle, WA). Plasmids encoding rapid cycling Arf mutants were generated by us as previously described (*Santy, 2002* and *Moravec et al., 2012*). GFP-STIM1, mCherry-STIM1 and YFP-VAP-A were gifts from Isabelle Derre (Univ. of Virginia, Charlottesville, VA). mCherry-CRY2-5-PtaseOCRL and CINB-CAAX were gifts from Olaf Idevall-Hagren (Uppsala University, Sweden) and the preparation of these reagents has been described in *Idevall-Hagren et al. (2012)*. Plasmids expressing probes for PI4P, P4M2X were gifts from Tamas Balla (NIH, Bethesda, Maryland). Plasmid expressing GFP-PLCδ was a gift from David Castle (Univ. Virginia, Charlottesville, VA). Plasmids expressing GFP- E-Syt1 were gifts from Pietro De Camilli (Yale Univ. New Haven, CT). GST fusion construct containing IQSec1 fragment 152–212 was prepared by cloning cDNA corresponding to codons 152–212 into pGEX-4T-1 using BamHI and EcoRI sites.

Two rounds of site-directed mutagenesis were used to generate an shRNA resistant form of IQSec1. The following primer pairs were used. Pair 1 F: GCGGAGTATGGGGGGCGCCTGG, R: TCGAGCATCTCCACCTGCTTGTCCTGC, Pair2: F: GCTCGAGCGGAAATACGGGGGGCGCC, R: A TCTCCACCTGCTTGTCCTGCAGGTCC The shRNA resistant IQSec1 plasmid was used as a template to generate the IQSec1 E606K catalytically inactive mutant and IQ mutant. The following primers were used for IQSec1 E606K mutant. Forward Primer: TGTCCAAGGGAAGGCTCAGAAAG and Reverse Primer: CGGATGTGCGCCTGGAAT and for IQ mutant, F: AACCGCATGTCACGCCGG, R: TACCAGGCGCCCCCCGTA. IQSec1$^{116-1114}$ was generated by sub-cloning this fragment between HindIII and EcoRI sites of pmCherry-C1. This plasmid was then used as a template to generate other mutants by site-directed mutagenesis.

| IQSec1 mutant | Forward primer | Reverse primer |
|---|---|---|
| IQSec1$^{116-870}$ | TAGCCCAGCATGTCCCAGTGC | CCGCACGACGCCTTTCTG |
| IQSec1$^{116-833}$ | TAGCAAGACCGGAAGAAATTCACCG | AGGGTTGGGGGCGTTGAA |
| IQSec1$^{116-759}$ | TAGCACCAGCGAGAAATCTTC | TAGTCCGAGTTTCTGGGG |
| IQSec1$^{116-502}$ | GGGACTCGCCTAGCCTTTAGCAACGATG | AGCTGTTGCGGGCCTCCT |
| IQSec1$^{116-245}$ | AGCCGCAGCCTGCACACTG | AGTTGAGGGCATCGTCGATGG |
| IQSec1$^{116-212}$ | TAGCCCCCTCCAGTGACTTT | CGGAGACTTGAGGGTGGT |
| WT- IQSec1 Δ$^{152-212}$ | GCCCCCTCCAGTGACTTT | GCGGTTCTCTGACATGGAG |

Site-directed mutagenesis was used to also generate an shRNA resistant form of ORP3. The following primer pair was used. F: CGCGATATCAGCGTATGCATCTAG and R: AACGCTGCCACATA TACCATCCTTTC.

## Knockdowns

Lentiviral vectors targeting IQSec1 were purchased from Sigma Aldrich, (St Louis, MI) and lentiviral particles were prepared as per the manufacturer's protocol. MDA-MB-231 cells were transduced with virus containing the shRNA for 24 hr, selected using 2 µg/ml of puromycin and analyzed 5 days later. The following sequences were used for knockdown: TRCN0000428512 (AGATGCTAGAACGAAAGTATG), TRCN0000419378 (CCTTCTCTAGGCAAGTGAAAT) TRCN0000433403 (ACTACTCAGACGGTGACAATG), TRCN0000149784 (CCAGTACCAGATGAACAAGAA) and TRCN0000180095 (CTTTGAGGTTCCAGACCCAAA). Lentiviral vectors targeting ORP3 were gifts from Johnny Ngsee (University of Ottawa, Canada). The following sequences were used for knockdown: TRCN0000158226 (CGTGGCATCATTTGGGAAGAA), TRCN0000157183 (GCCTTTGCCATA TCAGCGTAT), TRCN0000152169 (CAAAGTCTTTATTGCCACCTA), TRCN0000155275 (GC TGGAAGCAAGCCATTTAAT) and TRCN0000151606 (GAAGCGTAGCAGTATATCAAA) For Arf5 knockdown TRCN0000047990 (CAGATGCGGATTCTCATGGTT), TRCN0000047988 (GCTGCTGGTA TTTGCCAACAA), TRCN0000047989 (CCAACCATAGGCTTCAATGTA), TRCN0000047991 (GA TGCAGTGCTGCTGGTATTT) and TRCN0000286637 (GTCCAAGAATCTGCTGATGAA) and for Arf6 knockdown TRCN0000048003 (GTCAAGTTCAACGTATGGGAT) were used.

## RT-PCR analysis of IQSec1 splice variants

RNA was extracted using an RNeasy kit (Qiagen). Protoscript II (NEB, MA) first strand cDNA synthesis kit was used to prepare cDNA from 1 µg of RNA. To identify splice variants, the following primer pairs were used. PCR products were cloned into the TOPO vector and sequenced to validate their identity.

| Splice variant | Forward primer | Reverse primer |
|---|---|---|
| NP_001127854.1 | ATGGCTTGCAGAAGACGCTATTTCG | GAGATGGACACTCTCGTGTTGATGTAGC |
| NP_055684.3 | ATGTGGTGCCTGCACTGCAACTC | TTAGGAGCACAGCACTGACGGC |
| NP_001317548.1 | ATGCTAGAACGAAAGTATGGGGGGGC | CTAATGATGGACCCCGGCTTC |
| XP_011532610.1 | ATGTGGTGCCTGCACTGCAACTC | GAGATGGACACTCTCGTGTTGATGTAGC |
| XP_024309613.1 XP_011532616.1 | ATGCTAGAACGAAAGTATGGGGGGGC | GAGATGGACACTCTCGTGTTGATGTAGC |
| XP_011532609.1 | ATGGCTTGCAGAAGACGCTA TTTGAGCTCC | GAGATGGACACTCTCGTGTTGATGTAGC |
| XP_011532608.1 | ATGAAGGGGTATGGCAGTGCCGTG | GAGATGGACACTCTCGTGTTGATGTAGC |
| XP_011532607.1 | ATGCTCAAGTTCAAGGCCTT TTGCCTGG | GAGATGGACACTCTCGTGTTGATGTAGC |

## Real time PCR

Analysis was performed using the Applied Biosystems validated TaqMan primer-probe set (Hs00208333/spans exon 6 and 7) and the ABI PRISM SDS7000 sequence detection system (Applied Biosystems). The $\Delta CT$ method was used to quantify relative mRNA levels as described in the ABI user guide, 1997. 18 s RNA was used as reference and internal standard to compare IQSec1 mRNA levels for quantification.

## Drug treatments

For live cell experiments, cells were imaged for 150 s prior to addition of drug. Thapsigargin (1 µM) or PMA (2.5 µM) was added and cells were imaged live for additional 10–30 min. For biochemistry, cells were cultured in serum free media for 3 hr and then treated with thapsigargin (1 µM) for 30 mins or with PMA (2.5 µM) for 60 mins. For experiments with BTP2 (25 µM) or Gö6983 (20 µM), cells were pretreated with these drugs for 3 hr and then used for either live cell imaging or biochemistry.

## Co-precipitation studies

For protein mass spectrometry, HEK-293 cells expressing GFP-IQSec1 were lysed on ice with NP-40 lysis buffer (0.1% Nonidet P-40, Tris HCl pH7.4, 150 mM NaCl) containing a cocktail of protease inhibitors. Lysate were spun at 14000 rpm for five minutes at 4°C. Supernatant was collected and incubated with GFP-TRAP_A beads (Chromotek, Germany) for sixty minutes. Beads were washed four times with lysis buffer, boiled with SDS-PAGE sample buffer, resolved on 4–20% gradient gels and stained overnight with Gelcode blue stain (Thermo Fisher Scientific, Rockford, Il), followed by 4 to 5 washes 15 min each in water. Five bands were cut out spanning the length of the lane and used for protein mass spectrometry analysis.

To test protein-protein interaction, plasmids encoding tagged proteins were expressed in HEK-293 cells and cells were lysed on ice in NP-40 lysis buffer as described above. Lysates were spun at 14000 rpm for five minutes, supernatants collected and incubated with 1 µg antibody of choice (anti-mCherry or anti-GFP or anti-Xpress) for sixty minutes on rotation at 4°C, followed by an incubation with protein A beads (Cell Signaling, Danvers, MA) for thirty minutes. Beads were then washed four times with lysis buffer, boiled with SDS-PAGE sample buffer and subjected to western blotting.

## Protein mass spectrometry analysis

Protein gel samples were reduced and alkylated using DTT and iodoacetamide, respectively. Samples were digested overnight using trypsin (37°C) and de-salted the following day using solid phase extraction (SPE). LC-MS/MS experiments were performed on a Thermo Scientific EASY-nLC 1200 liquid chromatography system coupled to a Thermo Scientific Orbitrap Fusion Lumos mass

spectrometer. To generate MS/MS spectra, MS1 spectra were first acquired in the Orbitrap mass analyzer (resolution 120,000). Peptide precursor ions were then isolated and fragmented using high-energy collision-induced dissociation (HCD). The resulting MS/MS fragmentation spectra were acquired in the ion trap. MS/MS spectral data were searched using Proteome Discoverer 2.1 software (Thermo Scientific) against entries included in the Human Uniprot protein database. Search parameters included setting Carbamidomethylation of cysteine residues (+57.021 Da) as a static modification and oxidation of methionine (+15.995 Da) and acetylation of peptide N-termini (+42.011 Da) as dynamic modifications. The precursor ion mass tolerance was set to 10 ppm and the product ion mass tolerance was set to 0.6 Da for all searches. Peptide spectral matches were adjusted to a 1% false discovery rate (FDR) and additionally proteins were filtered to a 1% FDR.

## Identification of ORP3 lipid cargo

Immunoprecipitation of ORP3: For lipid mass spectrometry, HEK-293 cells expressing GFP-ORP3 were either left untreated or treated with thapsigargin for 30 min, and lysed on ice with NP-40 lysis buffer (0.05% Nonidet P-40, Tris HCl pH7.4, 150 mM NaCl) containing a cocktail of protease inhibitors. Lysates were spun at 14000 rpm for five minutes at 4°C. Supernatant was collected and incubated with GFP-TRAP_A beads (Chromotek, Germany) for sixty minutes. Beads were washed four times with lysis buffer. Samples were sent to the VCU Lipidomic and Metabolomics Core Facility (https://biochemistry.vcu.edu/Research/lipidomics_core.html) for analysis.

Extraction of Lipids: IP beads were transferred into 13 × 100 mm borosilicate tubes with a Teflon-lined cap (catalog #60827–453, VWR, West Chester, PA) by addition of 1 mL $CH_3OH$. Then 1 mL of $CH_3OH$ and 1 mL of $CHCl_3$ were added along with the internal standard cocktail (10 uL). The contents were dispersed using an ultra sonicator at room temperature for 30 s. This single-phase mixture was incubated at 48°C overnight. Debris was then pelleted in a centrifuge for 5 min at 5000 g, and the supernatant was transferred to a clean tube. The extract was reduced to dryness using a Speed Vac. The dried residue was reconstituted in 0.2 ml of the starting mobile phase solvent for untargeted analysis, sonicated for 15 s, then centrifuged for 5 min in a tabletop centrifuge before transfer of the clear supernatant to the autoinjector vial for analysis.

Untargeted analysis: Internal standards were purchased from Avanti Polar Lipids (Alabaster, AL) as their premixed SPLASH LIPIDOMIX mass spec standard. Internal standards were added to samples in 10 µL aliquots. Standards included 15:0-18:1(d7) PC, 15:0-18:1(d7) PE, 15:0-18:1(d7) PS, 15:0-18:1(d7) PG, 15:0-18:1(d7) PI, 15:0-18:1(d7) PA, 18:1(d7) LPC, 18:1(d7) LPE, 18:1(d7) Cholesterol Ester, 18:1(d7) MAG, 15:0-18:1(d7) DAG, 15:0-18:1(d7)−15:0 TAG, 18:1(d9) SM, and Cholesterol (d7).

For LC-MS/MS analyses, a Thermo Scientific Q Exactive HF Hybrid Quadrupole-Orbitrap Mass Spectrometer was used. The lipids were separated by reverse phase LC using a Thermo Scientific Accucore Vanquish C18+ 2.1 (i.d.) x 150 mm column with 1.5 µm particles. The UHPLC used a binary solvent system at a flow rate of 0.26 mL/min with a column oven set to 55°C. Prior to injection of the sample, the column was equilibrated for 2 min with a solvent mixture of 99% Mobile phase A1 ($CH_3CN/H_2O$, 50/50, v/v, with 5 mM ammonium formate and 0.1% formic acid) and 1% Mobile phase B1 ($CH_3CHOHCH_3/CH_3CN/H_2O$, 88/10/2, v/v/v, with 5 mM ammonium formate and 0.1% formic acid) and after sample injection (typically 10 µL), the A1/B1 ratio was maintained at 99/1 for 1.0 min, followed by a linear gradient to 35% B1 over 2.0 min, then a linear gradient to 60% B1 over 6 min, followed by a linear gradient to 100% B1 over 11 min., which held at 100% B1 for 5 min, followed by a 2.0 min gradient return to 99/1 A1/B1. The column was re-equilibrated with 99:1 A1/B1 for 2.0 min before the next run.

Each sample was injected two times for analysis in both positive and negative modes. For initial full scan MS (range 300 to 200 m/z) the resolution was set to 120,000 with a data-dependent $MS^2$ triggered for any analyte reaching 3e6 or above signal. Data-dependent $MS^2$ were collected at 30,000 resolutions. Data was analyzed using Thermo Scientific's Lipid Search 4.2 software.

## Immunoprecipitation of endogenous IQSec1

To precipitate endogenous IQSec1 we employed protocols previously described in *Hansen and Casanova (1994)*. Briefly, cells were lysed in 1 ml of 0.5% SDS lysis buffer (100 mM Tris pH 7.4, 150 mM NaCl, 0.5% SDS), and lysates were mixed with 0.5 ml of 2.5% Triton X-100 to sequester SDS.

Lysates were spun at 14000 rpm and supernatant was incubated with 2 µg of anti-IQSec1 antibody under rotation at 4°C for 60 min. Supernatant was then incubated with protein A sepharose beads for an additional 30 mins then washed with wash buffer (1% Triton X, 100 mM Tris pH 7.4, 150 mM NaCl). Beads were mixed with SDS-PAGE sample buffer, resolved on 4–20% gradient gels and immunoblotted with the same antibody.

## Pulldown experiments with GST-IQSec1[152-212]

HEK-293 cells expressing GFP-ORP3 or mCherry-ORP3 were lysed on ice with NP-40 lysis buffer as described above. Clarified supernatants were incubated for sixty minutes with beads containing 10 µg of GST alone or GST fused to IQSec1[152-212]. Beads were washed four times with lysis buffer, eluted with SDS-PAGE sample buffer and resolved on 4–20% gradient gels and immunoblotted with anti-GFP or anti-mCherry antibody.

For experiments involving λ-phosphatase, cells were lysed with lysis buffer containing 0.1% NP-40. The lysate was supplemented with $MnCl_2$ (1 mM) and incubated with 2000 Units of λ-phosphatase for 3 hr at 30°C. The phosphatase treated lysates were then incubated with GST-IQSec1[152-212] beads for sixty minutes. Beads were subsequently washed with lysis buffer containing 1% NP-40, boiled with SDS-PAGE sample buffer, resolved on 4–20% gradient gels and immunoblotted with anti-GFP or anti-mCherry antibody.

## GGA pulldown assays

Arf activation was measured using GST-GGA3-mediated pulldown, as previously described (*Santy and Casanova, 2001*). Briefly, cells expressing HA-tagged Arf5 were lysed on ice with chilled GGA-lysis buffer (1% NP-40, 25 mM Tris HCL pH7.4, 150 mM NaCl, 10% Glycerol, 5 mM $MgCl_2$). Clarified supernatants were incubated with GST-GGA3 beads for 30 min, washed three times in GGA-wash buffer (25 mM Tris HCL pH7.4, 150 mM NaCl, 10% Glycerol, 30 mM $MgCl_2$) and resolved on SDS-PAGE before immunoblotting with anti-HA antibody.

## Western blotting

Protein samples were resolved on 4–20% SDS-gradient gels (BioRad, Hercules, CA), then transferred to PVDF-FL membrane (Millipore Burlington, MA). Detection and quantification was carried out using a LI-COR Odyssey infrared scanning system using fluorescently labelled secondary antibodies. All western blots were scanned using the Odyssey Clx infrared imaging system from LICOR.

## Matrigel invasion assay

Biocoat matrigel invasion chambers from Corning (354480) were used for these assays. 100,000 cells, either mock depleted or depleted of IQSec1, Arf5 or ORP3 were loaded into the upper chamber in serum free media. Media containing 10% FBS was placed in the lower chamber and cells were allowed to invade for 6 hr at 37°C in a tissue culture incubator. Cells in the upper chamber were removed by gentle scrubbing using a cotton swab, followed by fixing cells that had migrated onto the lower surface with 4% PFA. Cells were stained with DAPI, imaged and counted. Three membranes were analyzed per group.

## Tumor spheroid assay

To generate tumor spheroids, we used approaches previously described by *Froehlich et al. (2016)*. Briefly, spheroid formation was induced by growing MDA-MB-231 cells (approximately 10,000 to 15,000 cells per well) in suspension in cell repellent surface 96 well plates (Greiner Bio-One, Germany) with complete media containing 3.5% matrigel for 48 hr. To prepare 3D collagen gels, 300 µl of bovine collagen (final concentration 1.5 mg/ml) in complete media (pH adjusted to 7.4 with 7.5% sodium bicarbonate) was pipetted into a 24 well TC treated plates and allowed to polymerize for 30 min at 37°C. Spheroids were mixed with 300 µl of collagen and then layered on top of collagen polymerized earlier and allowed to further polymerize for an additional 30 min at 37°C. Wells were subsequently topped with complete media and cells were imaged live at 5X magnification for 18–24 hr. For image analysis, the area of the spheroid at time 0 and after 18–24 hr was measured and data is presented as a ratio at these timepoints.

## Immunofluorescence microscopy

Cells were plated on fibronectin (5 µg/ml) coated coverslips. The next day cells were fixed with 4% paraformaldehyde for 10 min, washed with PBS and blocked in PBS containing 3% BSA (Bovine Serum Albumin) and 0.1% Triton X (PBST) for 30 min. Incubation with both primary and secondary antibodies was for 60 min at room temperature in blocking buffer. Coverslips were mounted on slides with Prolong Gold Antifade reagent (ThermoFisher Scientific, Rockford, Il).

## Confocal microscopy and image analysis

Images were captured after satisfying the Nyquist criteria for sampling using a 100X, 1.49 NA TIRF objective on a Nikon C1 Plus Confocal microscope. Z Sections of 0.25–0.5 microns were taken. For every experiment, a series of test images were taken to identify exposure gains that minimized over-saturation and this gain was subsequently used for all conditions. 12 bit images were analyzed using Nikon Elements software. A single optical section at the plasma membrane/matrix interface was analyzed per cell. Fluorescence intensities were calculated as mean fluorescence intensity.

## Live cell imaging using spinning disc confocal microscopy

Cells were plated on fibronectin coated (5 µg/ml), 35 mm glass-bottom dishes (Mat-Tek Corp). Imaging was carried out at 37°C approximately 18 to 24 hr after transfection. Before imaging, cells were transferred to pre-warmed imaging buffer (140 mM NaCl, 2.5 mM KCl, 2 mM $CaCl_2$, 1 mM $MgCl_2$, 20 mM HEPES, pH 7.4 also containing 10% FBS and 0.45% glucose). Cells were imaged using a 60X objective fitted to a Nikon TE 2000 microscope equipped with Yokogawa CSU 10 spinning disc and $512 \times 512$ Hamamatsu 9100 c-13 EM-BT camera. Green fluorescence was excited with a 488 nm/100 mW diode laser (Coherent) and collected by a BP 527/55 filter. Red fluorescence was excited with a 561 nm/100 mW diode laser (Coherent) and collected by a BP 615/70 filter. Multicolor images were acquired sequentially. Time lapse movies were captured at a typical frame rate of 15 s.

## Optogenetic depletion of PI(4,5)P$_2$

To deplete PM membrane pools of PI(4,5)P$_2$, we followed a procedure previously described in *Idevall-Hagren et al. (2012)*. Briefly, cells expressing GFP-ORP3, mCherry-CRY2-5-Ptase$_{OCRL}$ (a fusion protein containing the 5-phosphatase domain of OCRL attached to CRY2) and CINB-CAAX were exposed to blue light that induces dimerization of CRY2 and CINB, targeting the 5-phosphatase domain of OCRL to the PM. Blue light illumination was done by exposing cells to light from a 488 nm/100 mW diode laser for 10 frames (once every 15 s). PMA was added after 150 s and cells were imaged for an additional 30 mins.

For focal adhesion analysis, cells expressing mCherry CRY2-5 Ptase$_{OCRL,}$ CINB-CAAX and GFP-Paxillin were imaged every 15 s for 30 min. Exposing cells to blue light to image GFP-Paxillin also induced dimerization of CRY2 and CINB at the PM, depleting the PM of PI(4,5)P2.

## Measurement of focal adhesion assembly and disassembly rates

MDA-MB-231 cells were transfected with GFP-paxillin or DsRed paxillin and plated on fibronectin coated Mat-Tek dishes for 18–24 hr. Cells were imaged live using spinning disc confocal microscopy at 1 frame every 15 s for 60 min. Movies were compressed into a single grayscale stacked tiff file using Image J and uploaded to the focal adhesion server (https://faas.bme.unc.edu/) maintained by Shawn Gomez's lab at the University of North Carolina (Chapel Hill, NC). For assembly/disassembly rates, tracks with R-squared value greater than 0.8 were included in analysis. For further details on the server please refer to *Berginski et al. (2011)*.

## Measurement of PM occupancy

To measure PM occupancy, we used approaches previously described by *Giordano et al. (2013)*. Briefly, after applying a smoothing filter, areas with pixel intensities above background fluorescence were measured before (T$_0$) and after thapsigargin/PMA (T$_t$) treatment. The difference in areas at T$_t$ and T$_0$ was expressed as percentage of total cell area and plotted.

## Measurement of number of ORP3 puncta in cell protrusions and retractions

A double-blind study method was used to count the number of ORP3 puncta in cell protrusions and retractions. Briefly, cells were transfected with GFP-ORP3 and DsRed-paxillin, then imaged live every 15 s for 30 to 60 min. A protrusion was defined as an area where new focal adhesions form and mature, while a retraction was defined as an area where existing focal adhesions disassemble. Since paxillin was labelled in the red channel, protrusions and retraction were first identified in the red channel, a region of interest was drawn around them, followed by opening the green channel and counting the ORP3 puncta in the region of interest drawn. Two independent raters counted the ORP3 puncta and results were pooled and presented. To quantify the percentage of FA with ORP3 puncta, Imaris 9.5.1 software was used. Focal adhesions were identified using the surfaces function and puncta were identified using the spots function separately for protruding or retracting regions. Using the extension 'Find spots close to surfaces', puncta within a distance threshold of 1 µM to the nearest adhesion were identified and counted and the ratio of total associated puncta over total focal adhesions for each region was determined.

## Statistical analysis

Graphpad Prism7 was used for all statistical analysis. D'Agostino and Pearson's normality test was used to differentiate between parametric/non-parametric distributions. Mann Whitney's U test was used to test for significance between two non-parametric groups. In the case of multiple comparisons, the Kruskal-Wallis one-way analysis of variance was used along with Dunn's multiple comparison tests. For all other statistical analysis, student's 't' test was used to test for significance. For focal adhesion size and disassembly rates, data are presented as Tukey box-plots, where the top of the box represents the $3^{rd}$ quartile ($Q_3$) which captures the upper $75^{th}$ percentile and the bottom of the box represents the $1^{st}$ quartile ($Q_1$) which captures the lower $25^{th}$ percentile. The median is represented by the solid middle line. The upper whisker represents all values above the 75th percentile plus 1.5 times the IQR ($Q_3 + 1.5IQR$) and the lower whisker represents all values that fall below the 25th percentile minus 1.5 times the IQR ($Q_1-1.5IQR$). The interquartile range (IQR) is the difference between the third quartile and first quartile ($Q_3- Q_1$). The values plotted individually are outliers defined by Grubbs' test. In all figures, statistical significance between groups has been designated as: '*'$p<0.05$, '**'$p<0.001$, '***'$p<0.0001$, or 'NS' $p>0.05$.

## Acknowledgements

The authors thank David Castle, Bettina Winkler and Adam Greene for critically reading the manuscript, and Lloyd McMahon for help with image analysis. The authors are also grateful for DNA plasmids provided by Vesa Olkonnen, Rick Horwitz, Olaf Idevall-Hagren, David Castle, Isabelle Derre, Tamas Balla, Pietro De Camilli, Andrew Thorburn and John Ngsee. This work was supported by NIH grant (1RO1GM127361) to JEC.

## Additional information

### Competing interests

Kim Orth: Reviewing editor, *eLife*. The other authors declare that no competing interests exist.

### Funding

| Funder | Grant reference number | Author |
| --- | --- | --- |
| National Institute of General Medical Sciences | RO1GM127361 | James E Casanova |

The funders had no role in study design, data collection and interpretation, or the decision to submit the work for publication.

## Author contributions
Ryan S D'Souza, Conceptualization, Data curation, Formal analysis, Investigation, Visualization, Writing - original draft, Writing - review and editing; Jun Y Lim, Kelly Servage, Junmei Zhang, Nisha G Sosale, Matthew J Lazzara, Jeremy Allegood, Investigation, Methodology; Alper Turgut, Investigation; Kim Orth, Conceptualization, Writing - review and editing; James E Casanova, Conceptualization, Supervision, Funding acquisition, Project administration, Writing - review and editing

## Author ORCIDs
Kelly Servage (iD) http://orcid.org/0000-0001-7183-2865
Kim Orth (iD) http://orcid.org/0000-0002-0678-7620
James E Casanova (iD) https://orcid.org/0000-0002-0858-2899

## Decision letter and Author response
Decision letter https://doi.org/10.7554/eLife.54113.sa1
Author response https://doi.org/10.7554/eLife.54113.sa2

# Additional files
## Supplementary files
• Transparent reporting form

## Data availability
All data generated or analyzed during this study are included in the manuscript and supporting files.

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
