## [Decision Letter]

**Acceptance summary:**

Co-ordinated assembly and disassembly of integrin-mediated focal adhesions is essential for cell migration. Whilst Calcium influx has previously been shown to be a critical regulator of motility, the mechanisms are incompletely understood. This paper reports a new role for the lipid exchange protein ORP3 in this pathway, that acts as a spatial and mechanistic link between calcium influx at ER-plasma membrane contact sites and focal adhesion turnover. This dual action of ORP3 locally controls adhesion dynamics and is permissive for cell invasion.

**Decision letter after peer review:**

Thank you for submitting your article "Calcium-stimulated disassembly of focal adhesions mediated by an ORP3/IQSec1 complex" for consideration by *eLife*. Your article has been reviewed by three peer reviewers, one of whom is a member of our Board of Reviewing Editors, and the evaluation has been overseen by Anna Akhmanova as the Senior Editor. The reviewers have opted to remain anonymous.

The reviewers have discussed the reviews with one another and the Reviewing Editor has drafted this decision to help you prepare a revised submission.

Summary:

This manuscript reports a novel molecular complex regulating focal adhesion turnover and cancer cell invasion. The authors propose a Calcium-responsive mechanism centred around the lipid exchange protein ORP3. Calcium-influx-dependent PKC activation triggers cortical recruitment of ORP3 to ER-PM contact sites near focal adhesions. ORP3 is proposed to have a dual role, triggering activation of the small GTPase ARF5 by allosteric activation of the ARF GEF protein IQSec1, in addition to ORP3-mediated exchange of PI4P out of the PM, and exchange for a particular species of phosphatidylcholine. These dual pathways are proposed to control focal adhesion turnover. Overall, this work proposes a novel mechanism for the regulation of focal adhesion turnover. One general criticism is that the work brings together several concepts (GEF-ARF pair, lipid transfer, GEF-novel interactor complexes) without providing a clear and deep demonstration of how each stream controls focal adhesion function. In this sense, it is an amalgam of incomplete studies. Moreover, a pervasive issue is overstatement of conclusions, not supported by data. However, even if the details remain incomplete, this may be of general interest to the field.

Essential revisions:

1) A major concern is data replication and appropriateness of statistics. The extent of rigorousness of the data is ambiguous, making it difficult to judge the validity of a number of key data. The authors need to more clearly describe their analyses. Throughout the work, in figure legends, n values are given, sometimes in the thousands. Statistical analyses are performed on these values. However, such experiments are then listed as 5-8 cells per condition. Is this 5-8 cells per independent experiment? How many independent experiments? Essential to support the authors claims would be independently performed experiments with much more appropriate sample sizes. It is not appropriate to be comparing simply 5 cells in a single experiment to 5 in another.

2) With regard to mapping protein interactions using immunoprecipitations, single blots without quantitation are presented. It is stated that experiments were performed twice, with similar trends. The most appropriate would be to have at least triplicate experiments. In the absence of these, please include both independent experiments for all immunoprecipitations, plotted for the reader to see and judge conclusions.

3) In a number of places, 'rescue' experiments are performed to identify sufficiency of certain exogenously expressed proteins to restore a function lost upon RNAi to a certain protein. These are often referred to as restoring function to control levels. There is a consistent lack of support from the data that this occurs. For instance, in Figure 2B, a rapid-cycling mutant of Arf5 is proposed to rescue IQSEC1 depletion. Although the authors indicate that the metric used (focal adhesion disassembly) is decreased by IQSEC1 knockdown, then significantly increased by ARF5 rapid cycling mutant, this is modest at best. There is no indication or significance associated that this actually 'rescues' IQSEC1 loss to control levels. This is similarly the case for Figure 1E.

4) The data in Figure 3G and Supplementary Figure 3A do not convincingly show ORP3 or STIM1 is recruited to focal adhesions. The experiments would need to be performed using TIRF microscopy and full quantified to support this conclusion. Notably the appearance of ORP3 punctae at protruding membrane regions may be coincident with focal adhesions assembling, but not mechanistically linked. Many proteins accumulate in newly forming actin-based protrusions (not least because the volume of membrane increases at those sites); co-incidental data is not sufficient to support potential co-operation between these protein assemblies.

5) The binding of ORP3 to PC is interesting, but the study failed to link this finding back to focal adhesion formation; neither CA-Arf5 or ORF3 ORD mutants alter adhesion disassembly. The authors postulate both functions may be required for adhesion turnover, but this is not tested; and the possibility exists that neither are related to focal adhesion dynamics. In which case, it remains unclear as to the significance of the ORP3 lipid transfer with respect to adhesion dynamics. Moreover, it also remains possible/likely that the bulk effects on plasma membrane lipid partitioning indirectly alter the stability of numerous lipid-binding FA proteins, such as vinculin, kindlin and talin. It would strengthen the study to understand this, as well as the above potential role of Ras in integrin activation.

6) All experiments are performed with IQSEC1 'Isoform A'. However, this mutant only appears to partially rescue IQSEC1 knockdown. The authors indicate multiple isoforms of IQSEC1 expressed in MDA-MB-231. The data for this is missing. How did the authors detect both which 5’ end and which 3' end of the transcript is present to indicate select variants are present? Why was this isoform chosen? Do other variants 'rescue' fully?

7) The authors propose that ORP3 may dually a) activate IQSEC1 function and b) function to flip PI4P out of the PM and a specific PC into the OM. It is unclear whether these are the same mechanism or parallel. Depletion of PI4P by a PI4K inhibitor GSK-A1 inhibits invasion, suggesting that PI4P is needed, possibly as a substrate for ORP3 in migration. However, is PI4P needed for ORP3-stimulated IQSEC1-mediated ARF5 GTP-loading? Does GSK-A1 reduce ARF5-GTP levels?

8) Mass spec for interactors of IQSEC1, and lipid cargoes for ORP3 are identified and discussed. These data are not included, other than key targets. If the data is not present, the validity of claims for interaction or presence cannot be evaluated. These should not be included.

9) Figure 7G reports which lipid species may be exchanged by ORP3. However, replicate information is missing, as is any other species identity to put this in context of potential enrichment scores for this lipid.

10) The inclusion of PTPN13 and E-Syt3 as interactors of IQSEC1 is confusing (Figure 3). These are not further investigated. What is their reasoning for inclusion?

11) A previous paper (Weber-Boyvat et al., 2015) showed ORP3-VAPA binding regulates R-Ras to control integrin activation status. The authors cite this paper but fail to discuss these (highly relevant) findings in the context of their own study. Is it possible that all the effects seen here are mediated through this previously published pathway? It is notable here that the data in Figure 6—figure supplement 1 suggesting increased VAPA in the ORP3-IQSec1 complex was not convincing as shown and no statistics were presented. Nevertheless, if VAPA associates with this complex, this suggests Ras may also play a role. It would be very helpful to discuss this.

---

## [Author Response]

Essential revisions:1) A major concern is data replication and appropriateness of statistics. The extent of rigorousness of the data is ambiguous, making it difficult to judge the validity of a number of key data. The authors need to more clearly describe their analyses. Throughout the work, in figure legends, n values are given, sometimes in the thousands. Statistical analyses are performed on these values. However, such experiments are then listed as 5-8 cells per condition. Is this 5-8 cells per independent experiment? How many independent experiments? Essential to support the authors claims would be independently performed experiments with much more appropriate sample sizes. It is not appropriate to be comparing simply 5 cells in a single experiment to 5 in another.

We have increased the number of cells included in each analysis, and in each case have clarified the number of cells, the number of adhesions and the number of experiments (at least 3) in the corresponding figure legends. These new data have been incorporated into Figure 1C and E, Figure 2B and H, Figure 3C and E, Figure 4F, G and H, Figure 5A, B, C and E, Figure 6B and C, Figure 7J and K, Figure 2—figure supplement 1H and Figure 4—figure supplement 1D. In this regard, we note that there does not appear to be a standard in the field for what constitutes an appropriate sample size. A survey of the literature suggests a range from 3-12 cells and between 50 and 200 focal adhesions per condition: Juanes et al., JCB 2019 (200 FAs, 8-12 cells), Rajah et al. Sci Rep 2019 (10-20 FAs, 3-5 cells), Stehbens et al. Nat Cell Biol 2014 (50 FAs), Bouchet et al., *eLife* 2016 (12 adhesions, 5-6 cells per condition). Our expanded datasets now range from 12-30 cells per condition, spanning 3 separate experiments.

2) With regard to mapping protein interactions using immunoprecipitations, single blots without quantitation are presented. It is stated that experiments were performed twice, with similar trends. The most appropriate would be to have at least triplicate experiments. In the absence of these, please include both independent experiments for all immunoprecipitations, plotted for the reader to see and judge conclusions.

All immunoblots have now been quantified across 3 independent experiments, with the exception of Figure 6F and Figure 6—figure supplement 1E, for which duplicate experiments are shown in Figure 6—figure supplement 2B and Figure 6—figure supplement 1F respectively.

3) In a number of places, 'rescue' experiments are performed to identify sufficiency of certain exogenously expressed proteins to restore a function lost upon RNAi to a certain protein. These are often referred to as restoring function to control levels. There is a consistent lack of support from the data that this occurs. For instance, in Figure 2B, a rapid-cycling mutant of Arf5 is proposed to rescue IQSEC1 depletion. Although the authors indicate that the metric used (focal adhesion disassembly) is decreased by IQSEC1 knockdown, then significantly increased by ARF5 rapid cycling mutant, this is modest at best. There is no indication or significance associated that this actually 'rescues' IQSEC1 loss to control levels. This is similarly the case for Figure 1E.

For both Figure 1E and Figure 2B, we have increased the N for both cells and FAs measured for each condition, which has enhanced the differences between conditions. We also include measures of statistical significance for each pairwise comparison.

4) The data in Figure 3G and Supplementary Figure 3A do not convincingly show ORP3 or STIM1 is recruited to focal adhesions. The experiments would need to be performed using TIRF microscopy and full quantified to support this conclusion. Notably the appearance of ORP3 punctae at protruding membrane regions may be coincident with focal adhesions assembling, but not mechanistically linked. Many proteins accumulate in newly forming actin-based protrusions (not least because the volume of membrane increases at those sites); co-incidental data is not sufficient to support potential co-operation between these protein assemblies.

All images and videos provided were acquired using a spinning disc confocal microscope in the same focal plane as the adhesions. In our hands, images acquired using TIRF yielded nearly identical results, and while we appreciate the reviewer’s concern, we feel that re-doing all of the imaging and quantitation for every experiment would not affect the conclusions drawn. Also, the reviewer appears confused about where ORP3 puncta are accumulating – they are actually more abundant in disassembling, not newly forming adhesions. In our original manuscript we provided evidence that ORP3 puncta were roughly twice as abundant in retracting vs. protruding regions of the cell (Figure 3I). In this revision we also show that nearly 100% of disassembling adhesions had adjacent ORP3 puncta, while only 25% of newly formed adhesions did so (Figure 3J). This strongly suggests a preferential recruitment of ORP3 to ER/PM contact sites in response to local calcium signaling.

5) The binding of ORP3 to PC is interesting, but the study failed to link this finding back to focal adhesion formation; neither CA-Arf5 or ORF3 ORD mutants alter adhesion disassembly. The authors postulate both functions may be required for adhesion turnover, but this is not tested; and the possibility exists that neither are related to focal adhesion dynamics. In which case, it remains unclear as to the significance of the ORP3 lipid transfer with respect to adhesion dynamics. Moreover, it also remains possible/likely that the bulk effects on plasma membrane lipid partitioning indirectly alter the stability of numerous lipid-binding FA proteins, such as vinculin, kindlin and talin. It would strengthen the study to understand this, as well as the above potential role of Ras in integrin activation.

Our hypothesis that both Arf5 and ORP3 are required for adhesion turnover is based on the following data: (1) depletion of either Arf5 or ORP3 reduces the rate of adhesion turnover and significantly attenuates migration in 3D matrices (Figure 2B-C, Figure 3E-G); (2) ORP3-deficient cells are only partially rescued by expression of a lipid transfer-deficient mutant, suggesting that lipid transfer contributes to, but is not strictly essential for adhesion disassembly; and (3) CA-Arf5 can restore adhesion disassembly in IQSec1-depleted cells, but not in ORP3-depleted cells, suggesting that Arf5 is also necessary, but not sufficient for adhesion turnover. We agree that it is possible/likely that insertion of PC into the PM adjacent to adhesions may alter the stability of lipid-binding FA components, and have added text to this effect in our Discussion. However, it is unclear how we could test this experimentally; disassembling adhesions are by definition less stable, and are clearly more stable in the absence of ORP3 or Arf5.

Related to the topic of R-Ras and its association with ORP3; we have been unable to reproduce either the interaction between the two proteins or the effect of ORP3 knockdown on R-Ras activity previously described by the Olkkonen lab.

**Author response image 1. respfig1:** ORP3 fails to co-IP with HA-R-Ras (left), and depletion of neither ORP3 nor IQSec1 impacts R-Ras activity in a pulldown assay (**A**). Quantitation in panel **B** is from 3 independent experiments.

We also found that neither constitutively active nor dominant negative R-Ras interacted with ORP3 (not shown). All of these assays were performed using the conditions described by the Olkkonen lab. We have not included these data in the revised manuscript, but will do so if the reviewers deem it appropriate.

6) All experiments are performed with IQSEC1 'Isoform A'. However, this mutant only appears to partially rescue IQSEC1 knockdown. The authors indicate multiple isoforms of IQSEC1 expressed in MDA-MB-231. The data for this is missing. How did the authors detect both which 5-end and which 3' end of the transcript is present to indicate select variants are present? Why was this isoform chosen? Do other variants 'rescue' fully?

Using the sequences available at the NCBI, we designed PCR primers specific for the variable N- and C- termini, and sequenced the products generated after amplification from MDA-MB-231 mRNA. The corresponding primers and protocol are described in the Materials and methods. This analysis identified 3 isoforms, isoform A (NP_001127854.1), isoform X5 (XP_011532610.1) and isoform C (NP_001317548.1), also known as GEP100. In response to this reviewer’s concern, we tested the ability of all 3 detected isoforms to rescue IQSec1-depleted cells. As shown in the new Figure 1E, both isoform A and isoform C significantly restored adhesion turnover, while isoform X5 was less efficient. We also tested a fourth isoform (isoform B, NP_055684.3) that is not expressed in MDA-MB-231 cells, and it was statistically indistinguishable from a catalytically inactive mutant of isoform A. Because isoform A rescued most efficiently, that is the isoform we used for subsequent experiments.

7) The authors propose that ORP3 may dually (a) activate IQSEC1 function and (b) function to flip PI4P out of the PM and a specific PC into the OM. It is unclear whether these are the same mechanism or parallel. Depletion of PI4P by a PI4K inhibitor GSK-A1 inhibits invasion, suggesting that PI4P is needed, possibly as a substrate for ORP3 in migration. However, is PI4P needed for ORP3-stimulated IQSEC1-mediated ARF5 GTP-loading? Does GSK-A1 reduce ARF5-GTP levels?

We hypothesize that PI4P at the PM is indeed required as a substrate for exchange with PC from the ER, and that the lipid exchange mechanism is distinct from its allosteric activation of IQSec1. As noted by the reviewer, inhibition of this exchange inhibits migration. Interestingly, depletion of PI4P with GSK-A1 actually *increases* Arf5 activity (not shown). This suggests that ORP3 may be trapped in an open conformation that favors its interaction with IQSec1 if exchange cannot be completed. This finding complements our observation that rapid cycling Arf5 cannot drive adhesion turnover in the absence of ORP3 (Figure 7K).

8) Mass spec for interactors of IQSEC1, and lipid cargoes for ORP3 are identified and discussed. These data are not included, other than key targets. If the data is not present, the validity of claims for interaction or presence cannot be evaluated. These should not be included.

These data are now included as Source Data 1 and 2.

9) Figure 7G reports which lipid species may be exchanged by ORP3. However, replicate information is missing, as is any other species identity to put this in context of potential enrichment scores for this lipid.

We now include a new Figure 7I, which shows the level of enrichment for other lipids.

10) The inclusion of PTPN13 and E-Syt3 as interactors of IQSEC1 is confusing (Figure 3). These are not further investigated. What is their reasoning for inclusion?

We had originally included blots validating that PTPN13 and E-Syt3 interact with IQSec1 as additional confirmation of our mass spec analysis. However, we agree that this was confusing and have removed those blots.

11) A previous paper (Weber-Boyvat et al., 2015) showed ORP3-VAPA binding regulates R-Ras to control integrin activation status. The authors cite this paper but fail to discuss these (highly relevant) findings in the context of their own study. Is it possible that all the effects seen here are mediated through this previously published pathway? It is notable here that the data in Figure 6—figure supplement 1 suggesting increased VAPA in the ORP3-IQSec1 complex was not convincing as shown and no statistics were presented. Nevertheless, if VAPA associates with this complex, this suggests Ras may also play a role. It would be very helpful to discuss this.

As shown above (point #5), we have been unable to recapitulate either the interaction of R-Ras with ORP3 even in the presence of exogenous VAPA, or to detect any change in its activity upon knockdown of either ORP3 or IQSec1. For this reason, we have not included discussion of this point in the text.